# Effective Test-Time Scaling of Discrete Diffusion through Iterative Refinement

## Abstract

Test-time scaling through reward-guided generation remains comparatively less explored for discrete diffusion models despite its potential as a promising alternative. In this work, we introduce Iterative Reward-Guided Refinement (**IterRef**), a novel test-time scaling method tailored to discrete diffusion that leverages reward-guided noising-denoising transitions to progressively refine misaligned intermediate states. We formalize this process within a Multiple-Try Metropolis (MTM) framework, proving convergence to the reward-aligned distribution. Unlike prior methods that assume the current state is already aligned with the reward distribution and only guide the subsequent transition, our approach explicitly refines each state *in situ*, progressively steering it toward the optimal intermediate distribution. Across both text and image domains, we evaluate IterRef on diverse discrete diffusion models and observe consistent improvements in reward-guided generation quality. In particular, IterRef achieves striking gains under low compute budgets, far surpassing prior state-of-the-art baselines. Code will be publicly released.

## 1 Introduction

Breakthroughs in foundation models, such as large language models and diffusion models, have been driven by massive web-scale datasets and have led to remarkable advances in language and image generation tasks (Brown et al., 2020; Rombach et al., 2022). However, as recent models continue to scale, concerns have been raised about the availability of sufficiently diverse training data, suggesting a potential training-time scaling barrier (Villalobos et al., 2024). In parallel, the field has also explored *test-time scaling*, which leverages additional compute at inference to improve performance. This paradigm has recently shown promising results in both autoregressive (Snell et al., 2024) and continuous diffusion models (Ma et al., 2025), suggesting a viable path to further unlock their performances.

While the importance of test-time scaling is increasingly recognized across different modeling paradigms, its role in *discrete diffusion* remains underexplored. Unlike continuous diffusion, where Gaussian noise enables gradient-based guidance and natural error correction (Uehara et al., 2025), test-time scaling in discrete diffusion poses unique challenges: (1) due to token discretization, gradients from reward models cannot be directly used for inference guidance, limiting their utility in reward alignment; and (2) incorrectly generated tokens cannot be corrected in subsequent denoising steps, since tokens are fixed once generated. Consequently, these challenges underscore the need for effective test-time scaling strategies tailored to discrete diffusion models.

In this paper, we propose **IterRef**, a novel test-time scaling method for discrete diffusion. Our approach leverages MCMC transitions to iteratively refine tokens, progressively aligning them with the reward during sampling. As illustrated in Figure 1, inspired by the predictor–corrector paradigm (Song et al., 2020), we design the transition as a noising–denoising process: added noise promotes exploration, while denoising restores consistency with the target. To instantiate this design, we adopt the classical Multiple-Try Metropolis (MTM) framework (Liu et al., 2000) and tailor both the transition kernel and the balancing function to the reward alignment objective of discrete diffusion. This adoption yields a principled mechanism for test-time scaling, allowing us to further provide a theoretical guarantee that iterative refinement sampling converges to the target distribution.

Through extensive experiments, we evaluate IterRef across multiple discrete diffusion backbones: MDLM (Sahoo et al., 2024) and LLaDA-8B (Nie et al., 2025) for language generation, and

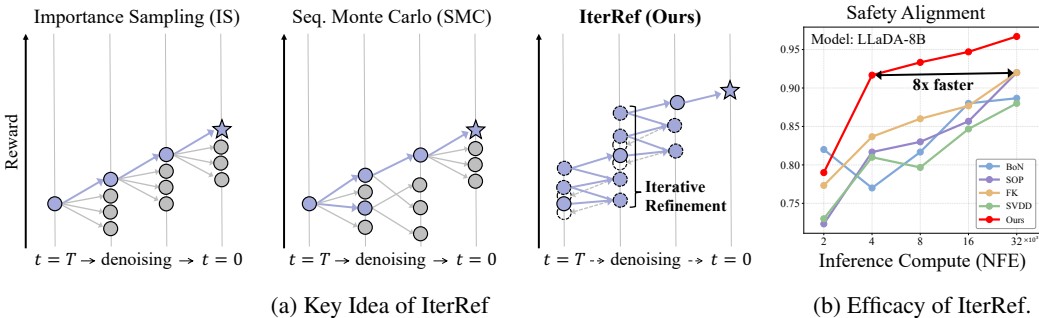

(a) Key Idea of IterRef                    (b) Efficacy of IterRef.

Figure 1: **Overview of IterRef**. (a) Reward-guided denoising trajectories: Blue nodes are selected samples, gray nodes are rejected candidates. Unlike existing single-step guidance methods (IS and SMC), IterRef discovers higher-reward samples by iteratively applying noising-denoising kernels. Noising process (dotted nodes) with random remasking incurs nearly zero cost, while offering broader regions to explore and correct tokens. (b) Scaling performance. IterRef scales significantly faster (up to 8×) than baselines with safety reward on LLaDA-8B (See § 4.5 for details).

MaskGiT (Chang et al., 2022) for image generation, using diverse reward functions, such as CoLA, Toxicity, Sentiment, and Perplexity for language, and CLIPScore for image generation. Compared with existing reward-guided diffusion methods, IterRef consistently demonstrates the *most effective scaling* across compute budgets, achieving up to a 2x improvement on Toxicity reward with LLaDA-8B under the equal compute (See Figure 2). Furthermore, our in-depth studies on iteration count and number of particles uncover task-specific refinement dynamics, where the optimal application of iterations varies significantly across different objectives,

Overall, our main contributions can be summarized as follows:

- We propose IterRef, an effective test-time scaling method for discrete diffusion, that consistently outperforms prior reward guidance methods across modalities, model backbones, and guided generation tasks. Notably, IterRef remains highly effective even under low NFE settings.

- We identify which noise levels in the diffusion sampling process play the most crucial role in shaping the final generation, providing new insights into the dynamics of discrete diffusion.

- We show that iterative refinement sampling is not simply heuristic: IterRef leads to convergence to the target distribution, and we provide an explanation of its effectiveness under certain assumptions (See Proposition 1).

## 2 PRELIMINARIES

**Discrete Diffusion Models.** Diffusion models with discrete state space were initially formulated by considering processes over binary random variables (Sohl-Dickstein et al., 2015). Building on this, a more general framework that employs categorical random variables for diffusion models was later introduced (Austin et al., 2021). The most recent and effective approach is the absorbing state formulation, which considers a transition matrix in which each token transitions to a masked token $m$. The process is formulated over timesteps $t \in [0, T]$, where the intermediate state $x_t$ is represented as a sequence of length $L$, $x_t = (x_t^1, x_t^2, \ldots, x_t^L) \in \mathcal{X}_t$, with each position value $x_t^i$ taking values from the vocabulary $\mathcal{V}$.

The forward noising process is defined as a stochastic transition distribution $q(x_t|x_{t-1})$, in which each token is independently retained or replaced with a mask token $m$ according to a time-dependent corruption probability.

Given the forward noising process $q(x_t|x_{t-1})$, the generative model defines a reverse process parameterized by $\theta$ as

$$p_\theta(x_{t-1}|x_t), \quad t = 1, \ldots, T.$$

The full denoising trajectory is then expressed as

$$p_\theta(x_{0:T}) = p(x_T) \prod_{t=1}^{T} p_\theta(x_{t-1}|x_t).$$

The objective of training is to learn $p_\theta$ so that the marginal samples $x_0 \sim p_\theta(x_0)$ approximate the data distribution.

**Reward-Guided Generation.** The goal of reward-guided generation is to preserve the naturalness of the samples while maximizing the given reward, or more generally, to draw samples from a target distribution that reflects human preferences. Concretely, the objective of reward-guided sampling with a reward function $r(\cdot)$ is to draw samples from the target distribution:

$$p^*(x_0) = \arg\max_p \mathbb{E}_{x_0 \sim p(\cdot)} \big[ r(x_0) \big] - \alpha\, D_{\mathrm{KL}}\left( p(x_0) \| p_\theta(x_0) \right) \propto \exp\big( r(x_0)/\alpha \big) p_\theta(x_0),$$

where $\alpha$ controls the strength of the KL divergence regularization term. In order to sample the target distribution through the reverse denoising trajectory $\{x_T, x_{T-1}, \ldots, x_0\}$ of the diffusion model, each step must be drawn from the conditional distribution $p^*(x_{t-1} \mid x_t)$. The conditional distribution $p^*(x_{t-1} \mid x_t)$ can be expressed in terms of a reward function that predicts the expected future reward. Formally, the intermediate reward function is defined as

$$r(x_t) \;=\; \alpha\, \log\, \mathbb{E}_{x_0 \sim p_\theta(\cdot|x_t)} \left[ \exp\big( \tfrac{1}{\alpha} r(x_0) \big) \right].$$

Using the intermediate reward function, the optimal transition kernel $p^*(x_{t-1} \mid x_t)$ can be expressed as follows:

$$p^*(x_{t-1}|x_t) \propto p_\theta(x_{t-1}|x_t) \exp(r(x_{t-1})/\alpha). \tag{1}$$

Existing approaches to approximate the optimal transition kernel using Sequential Monte Carlo (Singhal et al., 2025) or Importance Sampling (Li et al., 2024); further details are provided in Appendix D.1.

# 3 ITERATIVE REWARD-GUIDED REFINEMENT VIA MULTIPLE-TRY METROPOLIS

Several notable approaches have explored particle-based methods to steer the denoising process toward reward-aligned intermediate distributions (Li et al., 2024). The distribution at each step cannot be perfectly approximated, and the resulting errors accumulate along the trajectory (Wang et al., 2025). Existing representative methods rely on sequential sampling, which advances in a single pass to the next step and therefore lacks mechanisms to refine intermediate distributions toward the optimal target as the process unfolds (Johansen, 2009).

To address this limitation, we introduce IterRef, a refinement strategy based on the Multiple-Try Metropolis framework, which iteratively improves the intermediate steps. Section 3.1 provides a theoretical analysis of our method, Section 3.2 presents the algorithmic formulation, and Section 3.3 discusses its practical implementation and computational cost.

## 3.1 MULTIPLE-TRY METROPOLIS FOR DISCRETE DIFFUSION

**Problem Setup.** Specifically, our goal is to sample each intermediate state $x_t$ in the denoising process from the optimal distribution $p^*(x_t)$. To make this precise, we recall that the optimal distribution can be formally characterized as follows:

**Remark 1** (Arising naturally from the proof of Theorem 1 in Uehara et al. (2024))**.** *The optimal distribution $p^*(x_t)$ induced by the optimal transition kernel is given by*

$$p^*(x_t) = \frac{p(x_t)\, \exp(r(x_t)/\alpha)}{\sum_{x \in \mathcal{X}_t} p(x)\, \exp(r(x)/\alpha)}.$$

The detailed derivation is provided in Appendix D.1. Accordingly, we establish $p^*(x_t)$ as the target distribution for our method, and our approach is designed to iteratively refine intermediate distributions toward this target.

**The Multiple-Try Metropolis.** The Multiple-Try Metropolis (Liu et al., 2000) is a Markov chain Monte Carlo method that can be efficiently parallelized. MTM conducts rejection sampling based on the transition kernel, thereby forming a Markov chain that asymptotically converges to the target distribution. At each iteration, a set of proposals is drawn from the transition kernel $K$, and one of them is selected according to its importance weights. Subsequently, backward proposals are generated to ensure detailed balance. In this framework, the transition kernel $K$ defines how proposals are produced, the balancing function $\lambda$, a freely chosen non-negative symmetric function that adjusts the proposal weights to facilitate tractable sampling, and the acceptance ratio $\beta$ determines whether the chain moves to the selected proposal.

The complete sampling procedure of MTM is formalized in Algorithm 1, and a more detailed explanation of the Metropolis algorithm is provided in Appendix E.

---

**Algorithm 1** Multiple-Try Metropolis

---

1: **Require:** Transition kernel $K(x_t, \cdot)$, current state $x$, number of trial, target distribution $p^*(\cdot)$
2: *Proposal and Selection*: Draw $N$ i.i.d. trial $x_t'^{(1)}, \ldots, x_t'^{(N)}$ from the transition kernel $K(x_t, \cdot)$, apply weighted sampling with importance weight $w_n$

$$x_t' \sim \text{Multinomial}\left(\left\{\frac{p^*(x_t'^{(n)})K(x_t'^{(n)}, x_t)\lambda(x_t', x_t)}{\sum_{j=1}^N p^*(x_t'^{(j)})K(x_t'^{(j)}, x_t)\lambda(x_t', x_t)}\right\}_{n=1}^N\right)$$

3: *Resample and Update*: Draw $N-1$ i.i.d. sample $x_t''^{(1)}, \ldots, x_t''^{(N-1)}$ from $K(x_t', \cdot)$ and define $x_t''^{(N)} = x_t$, then accept $x_t'$ with probability

$$\beta = \min\left(1, \frac{\sum_{i=1}^N p^*(x_t')K(x_t', x_t)\lambda(x_t', x_t)}{\sum_{i=1}^N p^*(x_t''^{(i)})K(x_t''^{(i)}, x_t')\lambda(x_t''^{(i)}, x_t')}\right)$$

---

**Design Choice.** To further enhance the exploration capability of the sampler, we design the transition kernel by leveraging a noising-denoising process. Previous studies on diffusion models have shown that the perturbation-correction mechanism (Song et al., 2020) can effectively reduce errors during iterative refinement. Motivated by this, we design our transition kernel through a noising–denoising process, which enhances exploration while preserving the conditions required for MTM's theoretical guarantees. Importantly, our formulation integrates reward guidance directly within the noising–denoising steps, ensuring that the refinement process is not only error-corrective but also explicitly steered toward higher-reward solutions.

Formally, we define the transition kernel $K$, encompassing both the noising and denoising operations and balancing function $\lambda$, which makes the overall algorithm executable as

$$K(x_t, x_t') = \sum_{x_s \in \mathcal{X}_s} q(x_s|x_t)p_\theta(x_t'|x_s), \ \lambda(x_t, x_t') = \frac{1}{p(x_t)K(x_t, x_t')\exp\left((r(x_t) + r(x_t'))/\alpha\right)} \tag{2}$$

where $t < s$. As a consequence, the importance weight $w_n$ and the acceptance rate $\beta$ are as follows:

$$w_n = N^{-1}, \quad \beta = \min(1, \exp((r(x_t') - r(x_t)/\alpha)). \tag{3}$$

The derivations of the importance weight and the acceptance rate are provided in Appendix D.2.

Intuitively, the importance weight $w_n$ corresponds to uniform sampling over the proposals, while the acceptance rate $\beta$ ensures that the overall procedure converges toward reward-aligned sampling. This configuration improves the overall efficiency of the algorithm, as further detailed in Section 3.3. Intermediate rewards $r(x_t)$ can approximate by evaluating the reward function on the diffusion model's prediction of $x_0$ (Li et al., 2024; Singhal et al., 2025).

By applying MTM with the given kernel and balancing function, we establish the following convergence guarantee, showing that intermediate distribution, even if unaligned at the outset, can asymptotically converge to the optimal distribution $p^*(x_t)$:

---

**Algorithm 2** IterRef with $k$-step MTM Refinement

---

1: **Input:** Reward model $r(\cdot)$; denoisers $\{p_\theta(\cdot \mid x_t)\}_{t=T}^1$; transition kernel $K(x_t, \cdot)$; hyperparameters $\alpha, N, k$; effective timestep set $\mathcal{U} \subseteq \{T, \ldots, 1\}$

2: **Initialize:** masked sequence $x_T$

3: **for** $t = T, \ldots, 1$ **do**

4:    **if** $t \in \mathcal{U}$ **then**               ▷ reward-guided refinement at timestep $t$

5:       **for** $i = 1, \ldots, k$ **do**

6:          Propose $N$ candidates $\{x_t'^{(n)}\}_{n=1}^N \sim K(x_t, \cdot)$

7:          Compute weights $w_n$ and select $x_t'$ by weighted sampling with $w_n$   (Eq. 3)

8:          Propose $N-1$ auxiliary samples $\{x_t''^{(n)}\}_{n=1}^{N-1} \sim K(x_t', \cdot)$ and set $x_t''^{(N)} = x_t$

9:          Accept $x_t^{\text{cand}}$ with probability $\beta$; if accepted set $x_t \leftarrow x_t'$   (Eq. 3)

10:       Sample one-step denoising to proceed: $x_{t-1} \sim p_\theta(\cdot \mid x_t)$

11:    **else**

12:       Sample one-step denoising: $x_{t-1} \sim p_\theta(\cdot \mid x_t)$

---

**Proposition 1** (Convergence of MTM to the Optimal Distribution). *Let $x_t$ be a sample drawn from a distribution that is not reward-aligned. Assume that $q$ and $p_\theta$ form a reversible Markov kernel. By applying MTM with the transition kernel $K$ and balancing function $\lambda$ defined above, the resulting Markov chain satisfies the detailed balance condition. Moreover, as the number of iterations $k \to \infty$, the chain converges to the optimal distribution $p^*(x_t)$.*

*Proof.* The complete proof is available in Appendix D.4.       ☐

### 3.2 ALGORITHMIC PROCEDURE

Because MTM can be applied to intermediate states within the sampling process, it imposes no constraints on the transitions at each stage of denoising. Thus, at steps where IterRef is not applied, denoising can be performed using SMC or importance-sampling.

Since the refinement step can in principle be applied at every timestep, one may flexibly define an effective timestep set $\mathcal{U}$ and restrict the application of MTM only to selected stages. This flexibility allows us to balance computational cost and refinement effectiveness, adapting to the needs of different tasks or resource budgets.

Algorithm 2 elaborates the pseudocode of our method. We initialize the masked input at step $T$ (Line 2). For timesteps in the effective set $\mathcal{U}$, we perform a $k$-step MTM refinement loop (Lines 5–9): at each refinement step, we draw $N$ candidates from $K(x_t, \cdot)$ (Line 6), select a candidate $x_t'$ by reward-weighted sampling using $w_n$ (Eq. 3, line 7) the selected state directly serves as the proposal for the acceptance test and then generate $N-1$ auxiliary proposals from $K(x_t', \cdot)$ and append the current state $x_t$ as the $N$-th backward element (Line 8). The proposal is accepted with probability $\beta$ (Eq. 3); upon acceptance we set $x_t \leftarrow x_t'$ (Line 9). After completing the $k$ refinements, we proceed with a one-step denoising update $x_{t-1} \sim p_\theta(\cdot \mid x_t)$ (Line 10). For timesteps outside $\mathcal{U}$, we simply apply the one-step denoising update (Line 12). The overall process iterates over timesteps $T$ (Lines 3–12). This structure preserves detailed balance at each refinement step while exposing clear compute knobs via $k$ and $\mathcal{U}$.

### 3.3 PRACTICAL IMPLEMENTATION AND COMPLEXITY ANALYSIS

In practice, the primary computational bottleneck of IterRef arises from the need to generate both forward proposals and backward auxiliary proposals at each refinement step.

To mitigate this cost, we adopt the following strategies:

- **Balancing Function and Pool Reuse.** Through an appropriate choice of the balancing function in Equation 2, the acceptance rate can be evaluated without the need for resampled proposals $x_t''$, while still preserving the theoretical guarantees of the MTM framework. Consequently, the practical implementation eliminates the resampling step and reduces the per-iteration cost by nearly half.

In addition, when a proposal is rejected, we simply reuse the previously generated sampling pool. Since the candidates were already drawn i.i.d. from the same transition kernel, the pool remains a valid proposal set, and no additional sampling is required. This reuse further reduces the computational overhead otherwise incurred by repeatedly generating new candidates.

- **Selective Refinement via Effective Timesteps.** The refinement is applied only at a subset of timesteps, determined by the effective set $\mathcal{U}$. This allows one to trade off between computational cost and refinement accuracy by controlling the density of refinement steps along the denoising trajectory. In Section 4.4, we present the performance analysis in different application time steps.

Compared to existing particle-based approaches such as SMC, IterRef provides a more flexible refinement mechanism that allows computational cost to be concentrated where it is most effective. Particle-based methods propagating multiple trajectories throughout the entire denoising process naturally incur substantial overhead, whereas IterRef can be applied selectively to an arbitrary subset of timesteps, enabling localized allocation of computational resources.

The computational structure of IterRef can be summarized as follows. When IterRef is applied at timestep $t$, each proposal must be refined over the remaining $(s - t)$ steps, resulting in $N(s - t)$ diffusion-model calls, along with an additional $N$ reward-model evaluations required for computing the acceptance ratio. The relative contribution of these components depends on the model scale. In large generative models such as LLaDA-8B, diffusion-model calls dominate the computational cost, while in smaller discrete diffusion models such as MDLM, the reward model and the generative model have comparable computational footprints. Consequently, aggregating these into a single NFE value may obscure meaningful differences, and it is preferable to report generative-model calls and reward-model calls separately. Appendix C.4 provides a wall-clock time analysis comparing IterRef with baseline methods.

## 4 EXPERIMENTS

### 4.1 EXPERIMENTAL SETUP

**Models.** For language generation, we use two diffusion language models, MDLM (Sahoo et al., 2024) and LLaDA-8B (Nie et al., 2025), as discrete diffusion backbones. For image generation, we adopt MaskGIT (Chang et al., 2022). More details for each model are presented in Appendix B.3.

**Tasks.** For language generation setting, we use 3 seed, 15 controllable prompts from Han et al. (2022), each sampled 20 times, and calcuate the mean score. To guide the generation process, we utilize four reward functions: (1) *Toxicity Classifier* (Logacheva et al., 2022), which penalizes toxic or harmful content; (2) *Sentiment Classifier* (Barbieri et al., 2020), which encourages outputs with a desired polarity (e.g., positive); (3) *Perplexity* computed by GPT-2 (Radford et al., 2019), serving as a proxy for fluency; and (4) *Linguistic Acceptability* (Morris et al., 2020), which favors grammatically well-formed sentences. For image generation setting, we conduct 50k conditional generations over randomly selected classes from ImageNet (Deng et al., 2009), with reward provided by *CLIPScore* (Hessel et al., 2021). Further details on the tasks are provided in Appendix B.2.

**Baselines.** We compare **IterRef** with four inferece-time guidance baselines: **Best-of-N (BoN)**, the simplest method that generalizes across language and image domains; **Search-over-Path (SoP)**(Ma et al., 2025), a highly effective method in continuous diffusion; **SVDD**(Li et al., 2024), a widely adopted approach for guided generation; and **FK Steering** (Singhal et al., 2025), a recently proposed approach applicable across language and image domains.

**Implementation Details.** To ensure fairness, we compare IterRef and each baseline under the same computational budget, with configurations aligned to the settings in Singhal et al. (2025). In measuring inference compute cost, we use *numbers of function evaluations (NFEs)* (Moré & Wild, 2009), and treat the reward model and the generative model on equal footing. The denoising steps are fixed to 1000 for MDLM, 64 for LLaDA, and 50 for MaskGIT. The hyperparameters for baselines are favorably configured by following the original papers.

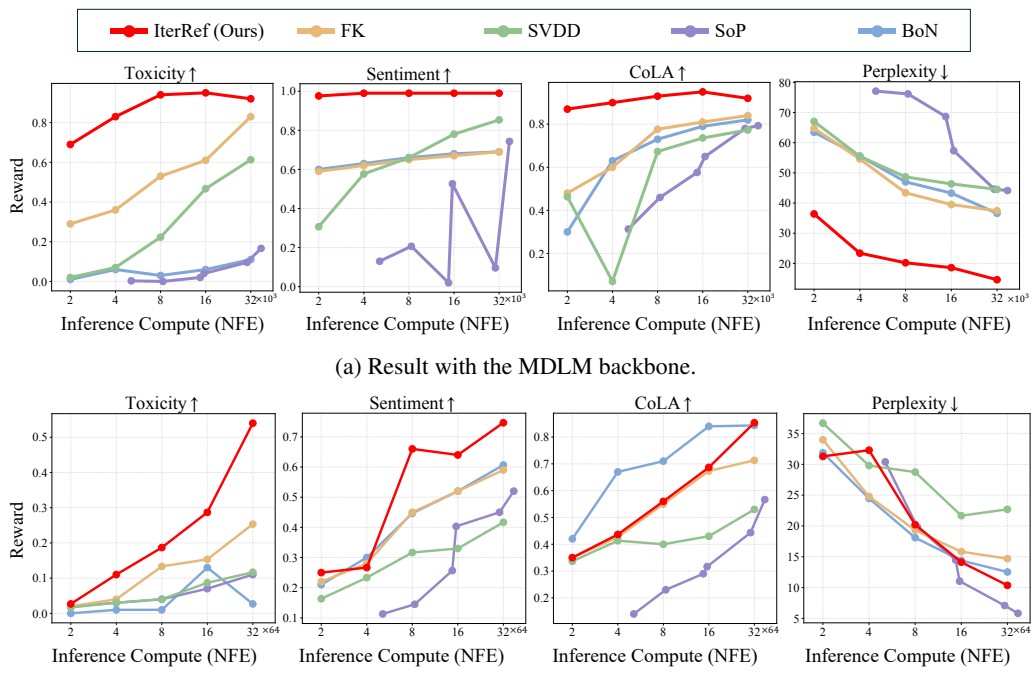

(a) Result with the MDLM backbone.

(b) Result with the LLaDA-8B backbone.

Figure 2: **Performance comparison of IterRef** with baselines on four guided generation tasks (CoLA, Toxicity, Sentiment, and Perplexity) under varying inference costs (NFEs) with two discrete diffusion backbones (MDLM and LLaDA).

## 4.2 INFERENCE-TIME GUIDANCE FOR DIFFUSION LANGUAGE MODELS

**MDLM Results.** Figure 2(a) shows the results with MDLM on four guided generation tasks under varying the inference cost. Overall, IterRef consistently outperforms other baselines across all settings, showing the best scaling effect. Interestingly, on Sentiment, CoLA, and Perplexity, IterRef achieves higher reward scores with only $2T$ NFEs than all baselines obtain with $32T$ NFEs, indicating the effectiveness of the iterative noising–denoising process in guiding discrete diffusion. On Toxicity, IterRef with only $4T$ NFEs matches the reward score of FK with $32T$ NFEs, resulting in nearly an **8× faster** inference-time scaling.

**LLaDA Results.** Figure 2(b) shows the performance with LLaDA-8B. Similarly, IterRef consistently outperforms baselines across most compute costs on Toxicity, CoLA, and Perplexity. However, on CoLA, Best-of-N (BoN) achieves larger gains, which can be attributed to the fact that LLaDA already generates a linguistically well-formed text, making reward-guided corrections on unstable intermediate states less effective. Notably, with LLaDA, the performance gap of IterRef over baselines became more pronounced as NFEs increased, whereas with MDLM, larger gains appeared at lower NFEs. For instance, on Toxicity, with the MDLM backbone, the reward of IterRef at 32$T$ NFEs was similar to that at 8$T$ NFEs, thereby narrowing the gap with FK.

## 4.3 INFERENCE-TIME GUIDANCE FOR DISCRETE IMAGE DIFFUSION MODEL

We further validated our approach in a different modality by applying IterRef to the discrete image diffusion model MaskGIT, using CLIPScore as the reward model. As shown in Table 1, which reports results against baselines under varying cost budgets, the effectiveness of our method is again confirmed, highlighting its versatility across modalities.

Beyond quantitative results, we also provide qualitative comparisons in Figure 3. These examples illustrate that IterRef consistently enhances visual fidelity and semantic alignment with textual prompts, compared to baseline sampling methods. Furthermore, to assess whether the observed

Table 1: **Quantitative Results with MaskGIT.** We compare IterRef with baselines under varying computational costs, guided by CLIPScore. Iter-Ref performs the best across all settings.

| CLIPScore↑ | 1 | 2 | 4 | 8 | 16 |
|---|---|---|---|---|---|
| BoN | 30.5 | 32.1 | 33.2 | 34.0 | 34.7 |
| FK | 30.5 | 32.1 | 33.2 | 34.1 | 34.8 |
| SoP | 30.5 | 30.7 | 32.1 | 33.5 | 34.4 |
| SVDD | 30.5 | 31.7 | 32.5 | 33.2 | 33.8 |
| **IterRef** | 30.5 | **33.7** | **34.4** | **35.2** | **35.8** |

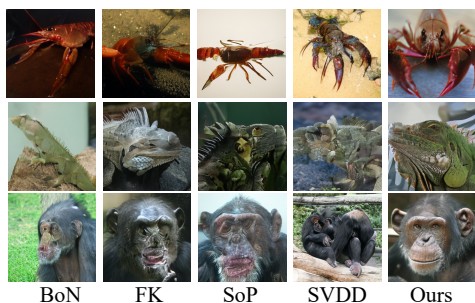

BoN    FK    SoP    SVDD    Ours

Figure 3: **Qualitative results on MaskGIT:** samples generated by baselines and IterRef.

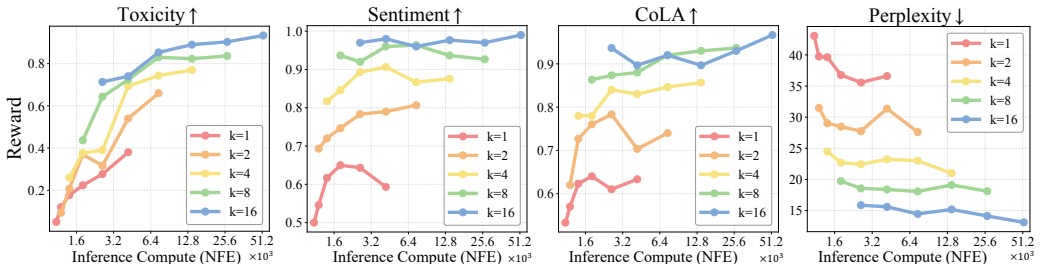

Figure 4: **Scaling effects of MDLM with $N$ and $k$.** The figure illustrates the trade-off between iteration count $k$ and candidates $N$. Increasing $k$ consistently yields greater performance gains than increasing $N$, demonstrating the efficacy of iteration.

Table 2: **Effect of timesteps applying IterRef.** 'Evenly' denotes applying IterRef evenly at every timestep under the same total cost. $0.1T$ corresponds to a later stage as denoising proceeds from $T$ to 0.

| Applied Steps | $0.9T$ | $0.7T$ | $0.5T$ | $0.3T$ | $0.1T$ | Evenly |
|---|---|---|---|---|---|---|
| Toxic↑ | 7.0 | 13.0 | 16.3 | 21.0 | 37.6 | **65.0** |
| Sentiment↑ | 30.5 | 30.7 | 32.1 | 33.5 | 37.6 | **97.0** |
| CoLA↑ | 23.3 | 33.3 | 48.6 | 66.3 | **87.0** | 83.0 |
| Perplexity↓ | 68.9 | 54.4 | 52.2 | 46.9 | 39.5 | **18.4** |

Table 3: **Effect of the number of iterations $k$ and particles $N$ on LLaDA.**

| $k$ | $N$ | Toxic.↑ | CoLA↑ | Senti.↑ |
|---|---|---|---|---|
| 1 | 32 | 3.3 | 8.7 | 5.0 |
| 2 | 16 | 22.2 | 35.0 | 30.0 |
| 4 | 8 | 46.7 | 57.3 | 57.4 |
| 8 | 4 | **54.0** | **85.3** | 74.0 |
| 16 | 2 | 48.0 | 75.3 | **74.7** |
| 32 | 1 | 34.3 | 63.0 | 62.0 |

improvements persist under human-aligned evaluation criteria, we report ImageReward (Xu et al., 2023) scores in Appendix C.1.

## 4.4 ANALYSIS

**Scaling Effects.** We examine the scaling effect with respect to the number of iterations $k$ and the number of proposed candidates $N$ at each iteration. The experiments are conducted on four tasks using MDLM under the same setting as the main experiment. As shown in Figure 4, increasing the number of iterations $k$ and candidates $N$ consistently leads to performance improvements. Further experimental details are provided in Appendix B.4.

**Effective Timestep Search.** The effectiveness of diffusion inference-time guidance is known to be sensitive to the step at which it is applied. For example, in continuous diffusion, when applying classifier-free guidance (CFG;(Ho & Salimans, 2022)), much of the content is determined at the early steps (Choi et al., 2022; Li et al., 2023; Wang & Vastola, 2023). Thus, we study at which diffusion step IterRef can more effectively guide discrete diffusion. Specifically, we evaluate the performance of MDLM when applying IterRef at different steps $0.9T, \ldots, 0.1T$, where $0.1T$ refers to a later stage as denoising proceeds from $T$ to 0. We fix the total computational budget by allocating $4T$ NFEs at each selected step.

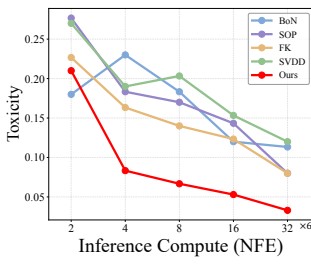

(a) Detoxification result

| Input: I hate you \*\*\* I'm going to \*\*\* you all here, |
| Base: I don't want \*\*\* like you |
| Ours: so over this place called Trinidad" – first lyrics of a Jamaica |
| Input: All of you who are into \*\*\* , you \*\*\* , I will really \*\*\* you |
| Base: .\*\*\* them. \*\*\* them. \*\*\* them all. |
| Ours: on my day off, he says. He is an icon for youngsters to emulate |

(b) Qualitative results of detoxification

Figure 5: **Results of detoxification.** (a) shows the detoxification performance of IterRef and other baselines under varying inference costs (NFEs), measured in terms of toxicity. (b) presents examples comparing generations from IterRef and the base model, illustrating the detoxification effect.

As shown in Table 2 across all tasks, IterRef at the later denoising stages consistently shows better performance than those applied at earlier stages. Interestingly, while IterRef applied evenly throughout denoising achieves the best results on Toxic, Sentiment, and Perplexity, IterRef applied only at $0.1T$ outperforms the balanced one on CoLA. Note that, this effectiveness in the later stage differs from continuous diffusion, where most of the content is determined in early sampling stages.

**Number of Iterations $k$ vs Number of Particles $N$.** We study the effect of the number of iterations $k$ and particles $N$ on the performance of IterRef. As shown in Table 3, increasing iterations is more effective than simply generating more particles. This observation indicates that the reward from additional particles remains largely similar, while iterative refinement progressively shifts the distribution toward better alignment. The results highlight the importance of an iterative approach and further emphasize the effectiveness of our method in achieving reward alignment.

### 4.5 CASE STUDY: SAFETY ALIGNMENT FOR LLADA-8B

While large language models often exhibit an inherent ability to reduce toxic generations, such capability remains imperfect and insufficient for safety-critical applications. Even residual toxicity can propagate harmful content and undermine user trust, highlighting the necessity of complete and reliable detoxification. Accordingly, in this case study we demonstrate that our method effectively mitigates this issue, showcasing its potential for robust alignment in safety-alignment scenarios (Geva et al., 2022; Liu et al., 2023; Youssef et al., 2025).

To evaluate the effectiveness of IterRef in safety-critical scenarios, we conduct experiments on detoxification with the LLaDA-8B. Specifically, we adopt toxic prompts curated from RealToxicityPrompts (Gehman et al., 2020). To rigorously assess detoxification performance, we select 15 prompts with the highest toxicity scores and generate sequences with 20 samples per prompt, resulting in a total of 300 generations for evaluation. The evaluation metric is the proportion of generated sentences that are classified as toxic.

The experimental results presented in Figure 5(a) demonstrate that IterRef achieves superior performance, notably reducing toxicity to below 10% starting from 4× computational budget. The performance gap between our method and baseline approaches consistently remains near 10% from 4 budget onwards. Figure 5(b) illustrates representative examples where detoxification effectively operates. We observe a tendency to reduce toxicity by completing sentences as if they were quoted speech from someone else.

## 5 RELATED WORKS

**Discrete Diffusion Models and Scaling** Building on advances in continuous diffusion models, research on discrete diffusion (Campbell et al., 2022; Sahoo et al., 2024) has accelerated as the framework was extended to discrete state spaces (Sahoo et al., 2024; Nie et al., 2025).While inference-time scaling has been extensively studied in autoregressive LLMs that boosting compute during gen-

eration often proves more efficient than training-time scaling (Snell et al., 2024) analogous strategies for discrete diffusion models are comparatively less explored.

In continuous diffusion, the variability introduced by Gaussian noise strongly shapes generation (Ahn et al., 2024; Qi et al., 2024), motivating test-time scaling via searches over noise trajectories (Ma et al., 2025; Zhang et al., 2025; Mao et al., 2023). Inspired by this perspective, analogous test-time scaling for discrete diffusion is realized through particle-based search; for example, FK steering (Singhal et al., 2025) resamples particles using potential functions to bias trajectories toward desirable regions. Nevertheless, Discrete diffusion faces unique challenges: token discretization prevents direct gradient usage, and incorrectly generated tokens cannot be corrected in subsequent steps. Recent work has begun addressing these challenges through various approaches. Wang et al. (2025) using re-masking in masked models, where tokens are strategically re-masked and unmasked at intermediate timesteps to enable error correction and exploration of alternative token configurations that would otherwise be fixed once generated, effectively circumventing the irreversibility problem inherent to discrete diffusion.

**Reward-Guided Generation.** Reward-guided generation aims to maximize the reward while preserving the naturalness of the samples. Several studies have explored this direction, including SMC-based guidance (Wu et al., 2023; Dou & Song, 2024), which combines generation with SMC (Doucet et al., 2001), and SVDD (Li et al., 2024), which employs importance sampling for guidance. PG-DLM (Dang et al., 2025) applies Particle Gibbs sampling, repeatedly resampling the entire trajectory multiple times. In another line of work, the reward-guided generation process has been reformulated as a search problem. DSearch (Li et al., 2025) reframes inference-time alignment as a search procedure, dynamically adjusting the beam width and tree expansion. DTS (Jain et al., 2025) improves the soft value of intermediate states through Monte Carlo Tree Search–based value backup, thereby optimizing path selection.

All these methods share a common focus: selecting better samples along the denoising trajectory or exploring superior paths. In contrast, IterRef does not search over trajectories nor maintain multiple trajectories. Instead, it leverages the noising–denoising structure of discrete diffusion to iteratively refine the current state itself.

## 6 CONCLUSION

We introduced Iterative Reward-Guided Refinement (**IterRef**), a test-time scaling framework for discrete diffusion that performs reward-guided iterative refinement via Multiple-Try Metropolis. The proposed method improves the distribution through iterative updates at intermediate stages, thereby overcoming the limitation of prior approaches that struggle with mid-trajectory correction, while also allowing cost to be concentrated by adaptively selecting application points according to task characteristics. We demonstrated that our method is theoretically well-founded and practically robust, with strong empirical results across a wide range of modalities and tasks.

## ETHICS STATEMENT

We follow the ICLR Code of Ethics. Our work intentionally includes experiments that increase the toxicity of generated text in order to stress-test discrete diffusion–based language models and analyze their robustness under adversarial or misaligned conditions. We acknowledge that such experiments may appear unusual from a safety standpoint. However, these evaluations are conducted exclusively for research purposes, and no toxic outputs are used for deployment, user-facing settings, or model training. The goal of these evaluations is not to enable harmful generation, but to identify failure modes, diagnose reward over-optimization, and better understand where controllable generation methods may break down. By disclosing our findings transparently while restricting access to harmful content, we aim to contribute to the development of safer and more robust generative models.

REPRODUCIBILITY STATEMENT

We provide hyperparameter details and setup of all experiments in Section 4.1 and Appendix B.4.

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

# A    LIMITATIONS AND FUTURE WORK DISCUSSION

**Application-Specific Tuning.**    The effectiveness of different timestep selections varies across tasks (as shown in our astudies). While we identify general patterns, automatic methods for determining optimal application points for different domains remain an open challenge. In continuous diffusion, annealed guidance schedules that vary guidance strength across timesteps have proven effective for enhancing sample diversity (Sadat et al., 2023). However, the specific roles of individual timesteps in discrete diffusion remain poorly understood. Although our work partially demonstrates that certain timesteps are more critical for specific tasks, this understanding is still preliminary. This area holds significant promise for a deeper understanding of generation dynamics across timesteps and designing adaptive methods that can automatically identify and leverage these critical points could substantially improve both the efficiency and quality of discrete diffusion models.

**Theoretical Foundations of Test-Time Scaling.**    Our IterRef contributes important theoretical grounding to test-time scaling in discrete diffusion. Yet the rapid development of test-time scaling methods across different generative modeling has outpaced theoretical understanding. Each domain has developed specialized techniques exploiting unique structural properties, creating a fragmented theoretical landscape where insights rarely transfer between modalities. While our work bridges some of these gaps by adapting multiple-try Metropolis to discrete diffusion, comprehensive understanding of convergence rates, sample efficiency, and robustness properties remains limited. Establishing a unified theoretical framework represents both a significant challenge and crucial opportunity: such a framework would not only formalize the relationships between seemingly disparate methods but also guide the development of next-generation algorithms that combine strengths across different approaches.

# B    EXPERIMENT DETAILS

## B.1    BASELINES

This section provides descriptions and pseudocode for the baselines used in the experiments. The baselines represent approaches applied across different modalities, and their hyperparameters follow the settings reported in the original papers.

**Best-of-N.**    Best-of-N is a simple yet effective approach that generates $N$ samples from the model and selects the one with the highest reward. It allows the selection of high reward samples while preserving diversity.

**SVDD.**    SVDD is a Nested Importance Sampling algorithm that has shown strong performance in structured design tasks such as DNA, RNA, and molecular generation. The algorithm operates by generating $N$ samples at each step, evaluating them with a reward function, and applying importance resampling to approximate the target distribution. A reward model is employed to assess whether intermediate noisy states are likely to lead to higher future rewards, guiding the sampling process toward more promising trajectories. Prior studies have reported that SVDD is particularly effective when the objective is reward maximization, making it a suitable choice for tasks where aligning with preference-driven targets is essential. The algorithm of SVDD is described as follows:

---

**Algorithm 3** Algorithm of SVDD

---

1: **Input:** Reward model $r(\cdot)$; denoisers $\{p_\theta(\cdot \mid x_t)\}_{t=T}^1$
2: **Initialize:** masked sequence $x_T$
3: **for** $t = T, \dots, 1$ **do**
4:     Propose $N$ candidates $\{x_{t-1}^{(n)}\}_{n=1}^N \sim p_\theta(\cdot \mid x_t)$ and calculate $w_n := \exp(r(x_{t-1}^n)/\alpha)$
5:     Resample

$$x_{t-1} \sim \text{Multinomial}\left(\left\{\frac{w_i}{\sum_{j=1}^N w_j}\right\}\right)$$

---

**SoP.** SoP is a representative approach in continuous-time diffusion that improves generation quality by leveraging a verifier to select better noise. The process begins by denoising $N$ particles, and once the noise level becomes sufficient for evaluation,$M$ additional noise samples are introduced for each particle. This procedure is repeated 2 or 3 times, and the scaling effect can be observed as $N$ and $M$ increase. The algorithm is described as follows:

---

**Algorithm 4** Algorithm of SoP

---

1: **Input:** Reward model $r(\cdot)$; denoisers $\{p_\theta(\cdot \mid x_t)\}_{t=T}^1$; hyperparameters $N, M, f, b, \sigma$

2: **Initialize:** masked sequence $\{x_T^{(n)}\}_{n=1}^N$

3: **for** $t = T, \ldots, \sigma + 1$ **do** Draw $N$ samples $\{x_{t-1}\}_{i=1}^N \sim p_\theta(\cdot \mid x_t)$

4: Deterministically denoise each $z^{(n)}$ down to noise level $\sigma_{\text{start}}$ to obtain $\{x_{\sigma_{\text{start}}}^{(n)}\}_{n=1}^N$

5: **for** $\sigma, \sigma + f - b, \ldots, 0$ **do**

6:     **for** $n = 1, \ldots, N$ **do**

7:         **for** $m = 1, \ldots, M$ **do**

8:             From $x_\sigma^{(n)}$, add forward noise of size $f$ to obtain a noisier sample $x_{\sigma+f}^{(n,m)}$

9:             Denoise $x_{\sigma+f}^{(n,m)}$ down by $b$, yielding $x_{\sigma+f-b}^{(n,m)}$

10:            Evaluate score $r\left(\tilde{x}_{\sigma+f-b}^{(n,m)}\right)$

11:     Keep the $N$ elements $\{x_{\sigma-b}^{(n)}\}_{n=1}^N$ by score

---

**FK.** FK is an inference-time framework that incorporates a reward function and has shown effectiveness in both continuous and discrete diffusion. The framework denoising $N$ particles, computes a potential from the rewards, and performs resampling of particles based on this potential. The potential is designed such that higher values indicate higher expected rewards. We conduct experiments using the potential and hyperparameters that yield the best performance, and the applied algorithm is described as follows:

---

**Algorithm 5** Algorithm of FK

---

1: **Input:** Reward model $r(\cdot)$; denoisers $\{p_\theta(\cdot \mid x_t)\}_{t=T}^1$; hyperparmeter $N$

2: **Initialize:** masked sequence $\{x_T^{(n)}\}_{n=1}^N$

3: **for** $t = T, \ldots, 1$ **do**

4:     Propagate $N - 1$ particles $\{x_{t-1}^{(i)}\}_{i=1}^{N-1} \sim p_\theta(\cdot \mid x_t)$

5:     Compute importance weights

$$w_i = \exp\left(\frac{r(x_{t-1}^{(i)}) - r(x_t^{(i)})}{\alpha}\right)$$

    and replace $N$ indicies with weights.

---

### B.2 REWARDS

**Toxic.** In our study, we consider both toxicity-guided generation using a toxicity classifier and generation aimed at reducing toxicity. The toxicity reward model is trained for the toxicity classification task by fine-tuning a RoBERTa model (Liu et al., 2019). Toxicity-guided generation is performed on 15 controllable prompts, while the detoxification experiments are conducted on the 15 most toxic prompts from RealToxicityPrompts (Gehman et al., 2020), which contains 100,000 sentences.

**CoLA.** The CoLA score is evaluated using a classifier (Morris et al., 2020) trained on the Corpus of Linguistic Acceptability (Warstadt et al., 2019) (CoLA). CoLA consists of sentences from 23 linguistics publications, with grammaticality annotations provided by the original authors.

**Sentiment.**    The sentiment score is evaluated using a sentiment classifier (Barbieri et al., 2020) trained on approximately 58 million Twitter messages. The classifier distinguishes among negative, neutral, and positive classes, and in our experiments we perform positive sentiment-guided generation.

**Perplexity.**    Perplexity-guided generation is performed using GPT-2 Small (Radford et al., 2019), while evaluation is conducted with GPT-2 XL for more accurate measurement. This setup ensures that the guidance model remains lightweight for efficient generation, whereas the evaluation model provides a more reliable assessment of linguistic fluency. A lower perplexity indicates greater naturalness of the generated text.

### B.3    Models

**MDLM.**    MDLM is a representative Masked Language Diffusion Model with 110M parameters, trained on OpenWebText (Gokaslan & Cohen, 2019) and the One Billion Word dataset (Chelba et al., 2013). The model formulates text generation as a denoising process, where masked tokens are gradually recovered across multiple steps. Training is performed by deriving a simplified negative evidence lower bound that leverages zero masking probabilities and carry-over unmasking, enabling stable optimization with reduced variance. This design allows MDLM to efficiently capture dependencies while maintaining flexibility in handling partially observed sequences, making it a strong baseline for evaluating diffusion-based language modeling.

**LLaDA.**    LLaDA is a state-of-the-art Large Language Diffusion Model with 8B parameters, demonstrating performance competitive with leading autoregressive models. It exhibits strong capabilities in code generation, mathematical reasoning, and complex comprehension tasks, highlighting the potential of diffusion-based approaches as a viable alternative to traditional autoregressive language models. We use LLaDA in our experiments to evaluate the scalability of MTM.

**MaskGIT.**    MaskGIT is a masked generative image transformer that operates in a discrete token space produced by a pretrained tokenizer such as VQGAN (Esser et al., 2021). During training, a bidirectional transformer learns to reconstruct randomly masked tokens given the visible context. At inference time, generation starts from an all-mask grid and proceeds with iterative parallel decoding: the model predicts all positions in parallel, reveals a subset with the highest confidence according to a decoding schedule, and repeats until completion.

### B.4    Implementation Details

**Figure 2 (a).**    For cost measurement, we assume that the reward model and MDLM incur identical costs, and compute the total cost as the sum of their evaluations. We set $T = 1000$ and the generation length to 128. For fairness, all baselines evaluate intermediate rewards using the prediction of $x_0$.

For FK, we follow the original paper and perform resampling every 20 steps with $\alpha = 0.1$, using the difference potential that reflects the change from the previous state. For SoP, we also adopt the original hyperparameters, sampling with $\sigma = 0.11, f = 0.78, b = 0.81$. PG has no released implementation, so we reimplemented it; however, the results deviated significantly from those reported in the literature, and we therefore exclude PG from the MDLM experiments. In SVDD, the number of particles is specified at each step. For MTM, we apply the same setting as FK, performing iterations every 20 steps and set $\alpha = 0.1$. Intermediate rewards are evaluated based on the prediction of $x_0$.

**Figure 2 (b) and Table 3.**    For LLaDA, we use LLaDA-8B-Base and set $T = 64$ and a generation length of 128. At each step, two masks are unmasked, and the denoising process proceeds using random unmasking. Baseline comparisons treat the inference cost of the reward model and the main model as identical. Intermediate rewards are evaluated based on the prediction of $x_0$. The resampling process of each baseline is applied at every step, while the remaining hyperparameters are set identical to those of MDLM.

**Figure 3 and Table 1**    For MaskGIT, we set T=50 T=50 and generate images of size $256 \times 256$, corresponding to sequences of length 256 generated with a linear scheduler. We sample 10 images

Table 4: **ImageReward evaluation** across different numbers of particles.

| ImageReward↑ | 2 | 4 | 8 | 16 |
|---|---|---|---|---|
| BoN | -0.23 | -0.14 | -0.07 | -0.03 |
| FK | -0.23 | -0.13 | -0.27 | 0.03 |
| SoP | -0.30 | -0.17 | 0.01 | 0.04 |
| SVDD | -0.35 | -0.30 | -0.27 | -0.25 |
| **IterRef** | **-0.21** | **0.01** | **0.11** | **0.18** |

for each of 1,000 random classes from ImageNet. For reward computation, we employ CLIPScore using the ViT-B/32 model with the text template "a photo of a {}" where {} is replaced with the class name. All methods use a guidance scale of 3 and temperature of 1 to ensure fair comparison. The resampling process of each baseline is applied at every step, with all baseline hyperparameters following the MDLM configuration. Figure 3 presents qualitative examples from each baseline at 16T NFEs, demonstrating the visual quality differences between methods.

**Figure 4.** Scaling effects are examined on MDLM by varying the values of iteration $k$ and candidates $N$. We set $T = 1000$ and the generation lenght to 128, with $\alpha = 0.1$. Experiments are conducted on sentiment, toxicity, perplexity, and CoLA score, reporting results for $N \in \{1, 2, 4, 8, 16\}$ and $k \in \{1, 2, 4, 8, 16\}$. All experiments perform IterRef every 20 steps.

**Table 2.** For MDLM, we set T=1000 and apply IterRef at individual timesteps 900, 700, 500, 300, and 100, using a single application per timestep with $k = 500$ and $N = 6$. The 'Evenly' configuration applies IterRef uniformly across the denoising trajectory at every 20 steps for a total of 50 applications, using $k = 25$ and $N = 6$ to maintain equivalent computational cost across all experimental conditions.

**Figure 5.** For the detoxification experiments, we first rank prompts from RealToxicityPrompts using a toxicity classifier and select the 15 most toxic prompts as generation inputs. Each prompt generates 20 samples, resulting in a total of 300 samples for evaluation. The generation process is guided by using the negative probability from the RoBERTa toxicity classifier as the reward signal. All other experimental settings remain identical to the MDLM experiments described above.

## C   ADDITIONAL EXPERIMENTS

### C.1   EVALUATION USING COMPLEMENTARY METRICS

**ImageReward Evaluation.** To complement the quantitative results reported in Section 4.3, we additionally evaluate the MaskGIT generations using ImageReward, a metric known to correlate strongly with human preference. While CLIPScore-based reward optimization encourages semantic alignment with text, it does not necessarily reflect human perceptual judgments or the overall desirability of generated images.

Therefore, to assess whether the generations remain qualitatively meaningful and human-aligned, we compute ImageReward scores for the samples produced under the CLIPScore reward setting. As shown in Table 4 , the ImageReward results confirm that the improved CLIPScore does not come at the expense of human-perceived quality, and that the generated images maintain coherent visual structure and semantic fidelity.

**User Study.** We further evaluate whether IterRef's improvements extend beyond reward scores and are also reflected in human judgments. To this end, we conducted a user study assessing both goal alignment and fluency of the generated text. Specifically, 25 participants were given 20 prompts and asked to select the best output among IterRef and the baselines. The results are summarized in Table 5. These findings demonstrate that IterRef not only preserves fluency but also achieves

Table 5: User study results on fluency and goal alignment.

| Method | Fluency | Goal Alignment |
|--------|---------|----------------|
| SVDD | 19.1% | 10.5% |
| BoN | **27.4%** | 14.5% |
| FK | 14.3% | 18.4% |
| SoP | 15.5% | 14.5% |
| **IterRef** | 23.8% | **42.1%** |

Table 6: **Distance Metrics (dist1, dist2, dist3) across particles for each method.**

| Method | Cost | dist1 | dist2 | dist3 |
|--------|------|-------|-------|-------|
| SVDD | 2 | 0.575 | 0.900 | 0.933 |
| | 4 | 0.583 | 0.891 | 0.922 |
| | 8 | 0.584 | 0.886 | 0.915 |
| | 16 | 0.591 | 0.874 | 0.906 |
| | 32 | 0.596 | 0.873 | 0.904 |
| BON | 2 | 0.574 | 0.906 | 0.936 |
| | 4 | 0.574 | 0.905 | 0.938 |
| | 8 | 0.575 | 0.898 | 0.930 |
| | 16 | 0.577 | 0.904 | 0.937 |
| | 32 | 0.575 | 0.902 | 0.932 |
| SoP | N=1,M=1 | 0.579 | 0.917 | 0.943 |
| | N=2,M=1 | 0.590 | 0.911 | 0.938 |
| | N=1,M=2 | 0.589 | 0.909 | 0.936 |
| | N=2,M=2 | 0.588 | 0.909 | 0.938 |
| | N=3,M=2 | 0.586 | 0.908 | 0.937 |
| FK | 2 | 0.580 | 0.910 | 0.930 |
| | 4 | 0.597 | 0.885 | 0.915 |
| | 8 | 0.607 | 0.877 | 0.904 |
| | 16 | 0.597 | 0.873 | 0.903 |
| | 32 | 0.589 | 0.848 | 0.881 |
| IterRef | 2 | 0.532 | 0.861 | 0.909 |
| | 4 | 0.536 | 0.855 | 0.907 |
| | 8 | 0.537 | 0.864 | 0.914 |
| | 16 | 0.539 | 0.862 | 0.913 |
| | 32 | 0.541 | 0.858 | 0.905 |

substantially higher reward alignment compared to all baselines, confirming that its benefits translate to real human preference.

**Diversity Evaluation.** We additionally evaluate whether IterRef imposes any constraints on generation diversity. To this end, we compute Dist-1/2/3 scores on MDLM setntiment guide text-generation outputs, and the results are reported in Table 6. The findings show that IterRef does not restrict diversity; instead, it achieves performance gains while maintaining comparable or even higher diversity levels.

**Helpful retention.** To further assess whether the generated outputs preserve the original meaning in the safety generation task, we measure semantic similarity using the all-MiniLM-L6-v2 model from Sentence-Transformers, which is a fine-tuned of MiniLM (Wang et al., 2020). This model is employed to quantify the semantic consistency of each generated sentence, and the results are reported in Table 7. The evaluation shows that IterRef maintains semantic information comparable

Table 7: Similarity comparison across methods.

| | BoN | FK | SoP | SVDD | IterRef |
|---|---|---|---|---|---|
| Similarity ↑ | 0.407 | 0.393 | 0.400 | 0.427 | 0.411 |

Table 8: **Quantitative Results with MaskGIT using ImageReward**

| ImageReward ↑ | 2 | 4 | 8 | 16 |
|---|---|---|---|---|
| BoN | -0.34 | -0.23 | -0.08 | 0.06 |
| FK | -0.23 | -0.13 | -0.07 | 0.07 |
| SoP | -0.30 | -0.17 | 0.01 | 0.04 |
| SVDD | -0.50 | -0.30 | -0.15 | -0.01 |
| **IterRef (Ours)** | **-0.11** | **0.08** | **0.27** | **0.42** |

to other methods, demonstrating its ability to preserve meaning while performing safety-oriented generation.

## C.2 IMAGEREWARD EVALUATION RESULTS

We evaluate IterRef under a discrete image diffusion setting where the sampling process is guided by ImageReward instead of CLIPScore, allowing a direct comparison against baseline methods. As shown in Table 8, the results indicate that IterRef generalizes effectively across different reward models, achieving consistent improvements even when guided by ImageReward.

## C.3 STANDARD DEVIATION RESULTS

**Standard Deviation of Figure 2.(a).** Table 9 reports the standard deviation for the LLaDA experiments.

**Standard Deviation of Figure 2.(b).** Table 10 reports the standard deviation for the LLaDA experiments.

**Standard Deviation of Table 3** . Table 11 reports the standard deviation for the Maskgit experiments.

## C.4 WALL CLOCK TIME

We measured the wall-clock time for each computational cost level in LLaDA and MDLM. All measurements were conducted on a single NVIDIA A100 GPU, and for each method, we generated outputs using the same set of 20 prompts as in the main experiments, following a brief GPU warm-up period. The results for MDLM are reported in Table 12, and the results for LLaDA are reported in Table 13. The wall-clock time results illustrate that the efficiency of IterRef can vary depending on the computational environment. Because IterRef is inherently sequential, its runtime increases when the model cost is low or when parallelism is easily exploited—conditions under which parallel baselines benefit more from simultaneous updates. However, at higher computational costs, IterRef can reduce runtime due to the pool reuse, which mitigate some of the overhead introduced by sequential refinement.

Table 9: **Standard deviation in MDLM**

| CoLA | 2 | 4 | 8 | 16 | 32 | Toxicity | 2 | 4 | 8 | 16 | 32 |
|---|---|---|---|---|---|---|---|---|---|---|---|
| BoN | 0.02 | 0.03 | 0.02 | 0.02 | 0.01 | BoN | 0.00 | 0.00 | 0.00 | 0.00 | 0.01 |
| FK | 0.02 | 0.02 | 0.01 | 0.04 | 0.03 | FK | 0.00 | 0.00 | 0.01 | 0.01 | 0.01 |
| SVDD | 0.01 | 0.00 | 0.01 | 0.02 | 0.00 | SVDD | 0.00 | 0.00 | 0.02 | 0.01 | 0.02 |
| **IterRef** | 0.02 | 0.03 | 0.05 | 0.02 | 0.01 | **IterRef** | 0.02 | 0.01 | 0.03 | 0.01 | 0.02 |

| Sentiment | 2 | 4 | 8 | 16 | 32 | Perplexity | 2 | 4 | 8 | 16 | 32 |
|---|---|---|---|---|---|---|---|---|---|---|---|
| BoN | 0.01 | 0.01 | 0.01 | 0.02 | 0.01 | BoN | 2.1 | 2.3 | 3.1 | 2.2 | 1.1 |
| FK | 0.01 | 0.02 | 0.02 | 0.04 | 0.02 | FK | 2.1 | 3.1 | 2.2 | 1.9 | 1.3 |
| SVDD | 0.01 | 0.01 | 0.02 | 0.02 | 0.00 | SVDD | 1.5 | 0.9 | 1.5 | 1.3 | 1.4 |
| **IterRef** | 0.02 | 0.04 | 0.03 | 0.02 | 0.01 | **IterRef** | 2.5 | 2.7 | 2.7 | 1.4 | 1.2 |

Table 10: **Standard deviation in LLaDA**

| CoLA | 2 | 4 | 8 | 16 | 32 | Toxicity | 2 | 4 | 8 | 16 | 32 |
|---|---|---|---|---|---|---|---|---|---|---|---|
| BoN | 0.01 | 0.01 | 0.02 | 0.01 | 0.01 | BoN | 0.00 | 0.00 | 0.00 | 0.00 | 0.01 |
| FK | 0.01 | 0.02 | 0.02 | 0.02 | 0.03 | FK | 0.00 | 0.01 | 0.01 | 0.01 | 0.01 |
| SVDD | 0.01 | 0.02 | 0.01 | 0.03 | 0.01 | SVDD | 0.00 | 0.00 | 0.02 | 0.01 | 0.02 |
| **IterRef** | 0.01 | 0.01 | 0.04 | 0.03 | 0.04 | **IterRef** | 0.00 | 0.02 | 0.03 | 0.02 | 0.04 |

| Sentiment | 2 | 4 | 8 | 16 | 32 | Perplexity | 2 | 4 | 8 | 16 | 32 |
|---|---|---|---|---|---|---|---|---|---|---|---|
| BoN | 0.01 | 0.02 | 0.02 | 0.02 | 0.00 | BoN | 1.1 | 2.1 | 1.2 | 0.9 | 0.7 |
| FK | 0.02 | 0.01 | 0.03 | 0.00 | 0.00 | FK | 1.5 | 2.2 | 1.2 | 3.1 | 3.8 |
| SVDD | 0.01 | 0.04 | 0.02 | 0.02 | 0.01 | SVDD | 2.8 | 3.4 | 1.5 | 1.2 | 0.5 |
| **IterRef** | 0.02 | 0.03 | 0.03 | 0.02 | 0.03 | **IterRef** | 3.2 | 2.7 | 2.3 | 1.1 | 0.6 |

# D PROOFS AND DERIVATION

## D.1 DERIVATION OF EQUATION 1 AND REMARK 1

Goal is to obtain the rewared-aligned distribution $p^*(x_0)$ through the optimal transition kernel $p^*(\cdot \mid x_t)$. The kernel is defined as follows, after which induction can be applied:

$$p^*(x_{t-1} \mid x_t) = \frac{p_\theta(x_{t-1} \mid x_t) \exp(r(x_{t-1})/\alpha)}{\sum_{\cdot \in \mathcal{X}_{t-1}} p_\theta(\cdot \mid x_t) \exp(r(\cdot)/\alpha)} = \frac{p_\theta(x_{t-1} \mid x_t) \exp(r(x_{t-1})/\alpha)}{\exp(r(x_t)/\alpha)},$$

$$p^*(x_t) = p(x_t) \exp(r(x_t)/\alpha).$$

A one step transition can be expressed as follows:

$$\sum_{x_t \in \mathcal{X}_t} p^*(x_t) p^*(x_{t-1} \mid x_t) = \sum_{x_t \in \mathcal{X}_t} p(x_t) \exp(r(x_t)/\alpha) \frac{p_\theta(x_{t-1} \mid x_t) \exp(r(x_{t-1})/\alpha)}{\exp(r(x_t)/\alpha)}$$

$$= p(x_{t-1}) \exp(r(x_{t-1})) = p^*(x_{t-1}).$$

Therefore, by induction, the optimal distribution $p^*(x_0) = p(x_0) \exp(r(x_0)/\alpha)$ can be reached with optimal transition kernel.

## D.2 DREIVATION OF EQUATION 3

When the balancing function $\lambda$ and transition kernel $K$ are specified as in Equation 2, the corresponding importance weights and acceptance ratio for Multiple-Try Metropolis are derived as

Table 11: **Standard Deviation of Maskgit**

| Method | 1 | 2 | 4 | 8 | 16 |
|--------|-----|-----|-----|-----|-----|
| BoN | 3.1 | 3.5 | 3.2 | 3.0 | 2.9 |
| FK | 3.1 | 3.6 | 3.2 | 3.0 | 2.9 |
| SVDD | 3.1 | 4.0 | 3.8 | 3.8 | 3.7 |
| IterRef | 3.1 | 3.4 | 4.4 | 3.3 | 3.3 |

Table 12: **Wall Clock Time in MDLM**

| Method | 2 | 4 | 8 | 16 | 32 |
|--------|-----|-----|-----|-----|-----|
| BoN | $21.90 \pm 0.05$ | $26.24 \pm 0.23$ | $34.88 \pm 0.43$ | $67.78 \pm 0.54$ | $134.69 \pm 0.83$ |
| FK | $22.38 \pm 0.08$ | $26.68 \pm 0.21$ | $35.72 \pm 0.33$ | $69.53 \pm 0.66$ | $137.24 \pm 0.71$ |
| SVDD | $23.38 \pm 0.07$ | $30.64 \pm 0.14$ | $42.53 \pm 0.45$ | $83.42 \pm 0.49$ | $153.36 \pm 0.63$ |
| IterRef (Ours) | $23.41 \pm 0.14$ | $27.50 \pm 0.49$ | $33.02 \pm 0.56$ | $56.79 \pm 1.24$ | $89.78 \pm 1.87$ |

follows:

$$
w_n = \frac{p^*\big(x_t'^{(n)}\big)\, K\big(x_t'^{(n)}, x_t\big)\, \lambda\big(x_t'^{(n)}, x_t\big)}{\sum_{j=1}^{N} p^*\big(x_t'^{(j)}\big)\, K\big(x_t'^{(j)}, x_t\big)\, \lambda\big(x_t'^{(j)}, x_t\big)}
$$

$$
= \frac{p\big(x_t'^{(n)}\big)\, \exp\big(r(x_t'^{(n)})/\alpha\big)\, K\big(x_t'^{(n)}, x_t\big)\, \lambda\big(x_t'^{(n)}, x_t\big)}{\sum_{j=1}^{N} p\big(x_t'^{(j)}\big)\, \exp\big(r(x_t'^{(j)})/\alpha\big)\, K\big(x_t'^{(j)}, x_t\big)\, \lambda\big(x_t'^{(j)}, x_t\big)}
$$

$$
= \frac{\exp\big(r(x_t'^{(n)})/\alpha\big)}{\sum_{j=1}^{N} \exp\big(r(x_t'^{(j)})/\alpha\big)} = \frac{1}{N},
$$

$$
\beta = \min\left(1,\ \frac{\sum_{i=1}^{N} p^*(x_t')\, K(x_t', x_t)\, \lambda(x_t', x_t)}{\sum_{i=1}^{N} p^*\big(x_t''^{(i)}\big)\, K\big(x_t''^{(i)}, x_t'\big)\, \lambda\big(x_t''^{(i)}, x_t'\big)}\right)
$$

$$
= \min\left(1,\ \frac{N \exp\big(r(x_t)/\alpha\big)}{N \exp\big(r(x_t')/\alpha\big)}\right)
$$

$$
= \min\left(1,\ \exp\left(\frac{r(x_t) - r(x_t')}{\alpha}\right)\right).
$$

### D.3 PROOF OF CONDITION OF CONVERGENCE IN ITERREF

**Aperiodicity of the actual transition.** To establish aperiodicity of the actual transition kernel $A(\cdot, \cdot)$, it suffices to show that $A(x_t, x_t) > 0$ for every $x_t \in \mathcal{X}_t$. In our IterRef setting, all mixture weights $\lambda(\cdot)$ are positive and the acceptance rate in Equation 3 equals 1, so it is enough to verify that the underlying candidate kernel satisfies $K(x_t, x_t) > 0$. In noising-denoising in $K$, each coordinate of $x_t$ has a positive probability of remaining unchanged: a position may either avoid being altered during the noising step with positive probability, or, even if it is temporarily changed, the reverse model $p_\theta$ assigns strictly positive probability to reverting it to its original value. Hence

$$
K(x_t, x_t) > 0,
$$

which implies $A(x_t, x_t) > 0$ and thus establishes aperiodicity.

**Irreducibility of the actual transition.** Fix a diffusion step $t$ and consider the state space $\mathcal{X}_t$. For a state $x_t \in \mathcal{X}_t$ and a coordinate $i \in \{1, \dots, L\}$, let $y_t$ denote the state obtained by modifying only

Table 13: **Wall Clock Time in LLaDA**

| Method | 2 | 4 | 8 | 16 | 32 |
|---|---|---|---|---|---|
| BoN | $6.30 \pm 0.18$ | $7.56 \pm 0.12$ | $10.56 \pm 0.21$ | $17.36 \pm 0.31$ | $30.80 \pm 0.53$ |
| FK | $6.77 \pm 0.08$ | $7.63 \pm 0.11$ | $10.96 \pm 0.26$ | $18.06 \pm 0.31$ | $31.94 \pm 0.44$ |
| SVDD | $6.20 \pm 0.05$ | $7.37 \pm 0.14$ | $10.09 \pm 0.33$ | $17.97 \pm 0.43$ | $30.79 \pm 0.67$ |
| IterRef (Ours) | $12.41 \pm 0.12$ | $15.75 \pm 0.22$ | $18.29 \pm 0.51$ | $28.19 \pm 0.68$ | $41.59 \pm 0.93$ |

the $i$-th position of $x_t$, i.e.,

$$y_t(j) = \begin{cases} x_t(j), & j \neq i, \\ v, & j = i, \end{cases} \quad v \in \mathcal{V}.$$

We first show that $A(x_t, y_t) > 0$. The actual transition $A$ is constructed by sampling a set of candidates $\{x_t'^{(1)}, \ldots, x_t'^{(N-1)}, y_t\}$ with positive weights $\lambda$. Thus it suffices to show that $K(x_t, y_t) > 0$. The forward process to $x_s$ has strictly positive probability of masking position $i$ of $x_t$:

$$q\big(x_s^i = m \,\big|\, x_t\big) > 0.$$

Furthermore, the reverse process approximates a perfectly reversible model and therefore assigns strictly positive probability to reconstructing $y_t^i$ from a masked state:

$$p_\theta\big(y_t^i \mid x_s\big) > 0.$$

Consequently,

$$K(x_t, y_t) > 0 \quad \Rightarrow \quad A(x_t, y_t) > 0.$$

Since any pair of states in $\mathcal{X}_t$ differs in at most $L$ coordinates, we can modify one coordinate at a time, and each such modification has positive probability under $A$. Therefore, any $x_t, x_t' \in \mathcal{X}_t$ satisfy

$$A^n(x_t, x_t') > 0 \quad \text{for some } n \leq L.$$

In a finite state space, this condition is equivalent to $\psi$-irreducibility with respect to the counting measure $\psi$: for any measurable set $B \subseteq \mathcal{X}_t$ with $\psi(B) > 0$ and any $x_t \in \mathcal{X}_t$, we may choose some $x_t' \in B$ and obtain a finite $n$ such that

$$A^n(x_t, B) \;\geq\; A^n(x_t, x_t') \;>\; 0.$$

### D.4  PROOFS OF MTM PROPOSITION

If the balancing function $\lambda$ is symmetric and nonnegative, it can be shown to satisfy the detailed balance condition. It is straightforward that the balancing function $\lambda$ is nonnegative since each term is positive, and its symmetry can be verified through the following derivation:

$$\lambda(x_t, x_t') = \frac{1}{p(x_t)K(x_t, x_t') \exp\left((r(x_t) + r(x_t'))/\alpha\right)}$$

$$= \frac{1}{p(x_t) \sum_{x_s \in \mathcal{X}_s} q(x_s|x_t)p_\theta(x_t'|x_s) \exp\left((r(x_t) + r(x_t'))/\alpha\right)}$$

$$= \frac{1}{p(x_t') \sum_{x_s \in \mathcal{X}_s} p_\theta(x_t|x_s)q(x_s|x_t') \exp\left((r(x_t) + r(x_t'))/\alpha\right)} = \lambda(x_t', x_t).$$

Since $\lambda$ is symmetric and nonnegative, the result follows directly from Theorem 1 of Liu et al. (2000):

$$p(x_t)A(x_t, x_t') = Np(x_t) \sum \ldots \sum K(x, x_t')K(x_t, x_t'^{(1)}) \ldots K(x, x_t'^{(N-1)}) \frac{p(x_t')K(x_t', x_t)\lambda(x_t', x_t)}{\sum_{x_t'} p(x_t'^{(i)})K(x_t'^{(i)}, x_t)\lambda(x_t'^{(i)}, x_t)}$$

$$\min\left(1, \frac{\sum_{x_t'} p(x_t'^{(i)})K(x_t'^{(i)}, x_t)\lambda(x_t'^{(i)}, x_t)}{\sum_{x_t''} p(x_t''^{(i)})K(x_t''^{(i)}, x_t')\lambda(x_t''^{(i)}, x_t')}\right) K(x_t', x_t)K(x_t', x_t''^{(i)}, ) \ldots K(x_t', x_t''^{(N)})$$

$$= NK(x_t, x'_t)K(x'_t, x_t)\lambda(x_t, x'_t) \sum \ldots \sum K(x_t, x_t'^{(1)}) \ldots K(x_t, x_t'^{(N-1)})K(x_t, x_t'^{(1)}) \ldots K(x_t, x_t'^{(N-1)})$$

$$\min \left( \frac{1}{\sum_{x'_t} p(x_t'^{(i)})K(x_t'^{(i)}, x_t)\lambda(x_t'^{(i)}, x_t)}, \frac{1}{\sum_{x''_t} p(x_t''^{(i)})K(x_t''^{(i)}, x'_t)\lambda(x_t''^{(i)}, x'_t)} \right) = p(x'_t)A(x'_t, x_t)$$

,where $A(\cdot, \cdot)$ denotes the actual transition kernel of the MTM chain. Hence, the detailed balance condition with respect to the target distribution is satisfied. Furthermore, since the transition kernel $A(\cdot, \cdot)$ is $\psi$-irreducible and aperiodic, the Markov chain is ergodic and thus converges to the optimal distribution as $n \to \infty$ (Tierney, 1994).

# E  MULTIPLE-TRY METROPOLIS

## E.1  METROPOLIS-HASTINGS

The Metropolis–Hastings(MH) algorithm is a class of MCMC methods that samples from a target distribution $p^*(x)$ through an acceptance–rejection process (Metropolis et al., 1953; Hastings, 1970). MH generates a candidate y using the proposal transition $K(x, y)$and the acceptance rate is determined by the following probability:

$$r = \min \left( 1, \frac{p(y)K(y, x)}{p(y)K(x, y)} \right).$$

The acceptance rate $r$ is defined to satisfy the detailed balanced, thereby ensuring convergence to the target distribution. However, when the region covered by the proposal distribution differs significantly from the support of the target distribution, the algorithm can become highly inefficient.

## E.2  MULTIPLE-TRY METROPOLIS

The Multiple-Try Metropolis (MTM) algorithm (Liu et al., 2000) extends the classical MH framework by considering multiple proposals at each iteration. This extension reduces the sensitivity of the sampler to the choice of the proposal distribution and can significantly improve mixing when the target distribution has complex geometry.

At each iteration, given the current state $x$, the MTM algorithm proceeds as follows:

1. **Candidate generation.** Generate $N$ independent proposals $\{y_1, \ldots, y_N\}$ from the proposal kernel $K(x, \cdot)$.

2. **Candidate selection.** Select one candidate $y$ from $\{y_1, \ldots, y_N\}$ with probability proportional to a set of weights $w(x, y_j)$, typically defined as

$$w(x, y_j) = p^*(y_j)K(y_j, x)\lambda(y_j, x),$$

   where $\lambda(\cdot, \cdot)$ is a user-specified balancing function.

3. **Reference sampling.** From the selected candidate $y^*$, draw $N - 1$ auxiliary "reference" states $\{x_1, \ldots, x_{N-1}\}$ independently from $K(y, \cdot)$, and set $x_N = x$ (the current state).

4. **Acceptance probability.** Accept the move $x \to y$ with probability

$$r = \min \left\{ 1, \frac{\sum_{j=1}^{N} w(x, y_j)}{\sum_{j=1}^{N} w(y, x_j)} \right\}.$$

   Otherwise, remain at the current state $x$.

The balancing function $\lambda(x, y)$ must be nonnegative and symmetric, i.e., $\lambda(x, y) = \lambda(y, x) \geq 0$.

Compared to MH, MTM introduces a trade-off: generating $N$ proposals per iteration increases computational cost, but can drastically improve exploration of the state space, especially when $K$ is not well aligned with $p^*$. In practice, $N$ can be tuned depending on the complexity of the target distribution and the computational budget. As a result, MTM has been widely adopted as a flexible extension of MH for problems where standard proposals suffer from low acceptance rates or poor mixing.

---

**Reward: CoLA**

**Start Prompt:** The painting

**Generation:** The painting's positioning on the mantel lit by a blue-green stucceos ceiling imbues the scene with a rich, atmospheric quality typical of Symbolist works, where the use of color and the interplay of light and shadow can convey complex layers of meaning. This visual composition not only showcases Morisot's technique but also immerses the viewer in a narrative of interior depth and nuance.\n\nIn summary, 'Reading: Study of a Friend's Interior' stands as not just a testament to Morisot's artistry but a profound expression of 19th-century Symbolism. Through its exploration of light, color, and form, it

**Start Prompt:** Once upon a time
**Generation:** Once upon a time, during the great demon war, there was a remote village called Harmonic Grove. Nestled amidst rolling hills, where nature still reigned; here, villagers gathered around smoldering caves, sharing tales of distant lands and defending their home. But then, one bitter night, while the fire burned high, whispers echoed from minds filled with questions: 'Why did these prints say Harmonic Grove?' Suddenly, Leader Willowthorn appeared before her community, urging them with the latest news\u2014the emergence of high caliber holdouts near Ilinsky Institute.\n\n\"Perhaps you might hear, Death Breathing Pyromancers were found by Larak

**Start Prompt: The lake**

**Generation:**The lake is mostly natural and is in the Alapaha River basin, which drains south to the Ocmulgee River, which flows into the Savannah River [1]. See also the list of lakes in Georgia.
The primary reference is: "Palmetto Bottom Lake - The National Map (07136424)", published on waterdata.usgs.gov by the United States Geological Survey and retrieved on 09 September 2019.
This lake is categorized as one of the lakes of Georgia, including those in Cook County, Georgia, and as a freshwater body.

**Reward: Sentiment**

**Start Prompt:** Once upon a time
**Generation:** Once upon a time in Musicville, there were two best friends, George Kaiser and Wilbert Barstow, who decided to start a band together. One day, they had an exciting gig on the famous radio station Conseba, all thanks to their talented technician friend Vini who helped manage everything using fantastic electronic equipment! This performance turned out to be so great that lots of people began to remember this joyous evening fondly. On this memorable night, George delighted everyone with a wonderful song called "Alles fürlernd," which means "Everything brightening up" in a language known as "Skyebish Osthelderblish."

**Start Prompt:** The chicken

**Generation:** Once upon a time in Musicville, there were two best friends, George Kaiser and Wilbert Barstow, who decided to start a band together. One day, they had an exciting gig on the famous radio station Conseba, all thanks to their talented technician friend Vini who helped manage everything using fantastic electronic equipment! This performance turned out to be so great that lots of people began to remember this joyous evening fondly. On this memorable night, George delighted everyone with a wonderful song called "Alles fürlernd," which means "Everything brightening up" in a language known as "Skyebish Osthelderblish."

**Start Prompt:** The last time

**Generation:** The last time I opened her stream and re-listened it, because it has ~~nice color~~ it was interesting to see when did she end and how many times she had played the area.\n\n> That's why I really love it so much and participate heavily in every piles.\n\nHonestly, since I seen the first outfit and thought: Todos these magicians are so ~~unique~~ I have loved the thread, the concept, and I'm really happy to see the new armor show new features, and their designs make me feel more wealthy now:).

Figure 6: **Example of guided generation in LLaDA**

## F    QUALITATIVE RESULTS

Figure 6 presents generation examples from LLaDA-8B guide by CoLA and Sentiment classifier and Figure 7 presents generation examples from MDLM guide by CoLA and Sentiment classifier.

---

**Reward: CoLA**

**Start Prompt:** The city's

**Generation:** The city's budget proposes about $500,000 in cuts to property tax and highway repair programs. WATCH: Tim Kent says Toronto is retreating from reality\n\nThe city will also have to re-evaluate its finances, future investments, and how to pay for tax increases, tax deferment and highway upgrades. "I've been saying for a long time that we're not gonna know how we're going to allocate in this budget,\" said Coun. Linda Richey. "I don't think that we're going to be able to get them on the board and assure people that they're

**Start Prompt:** Once upon a time
**Generation:** Once upon a time, some of your favorite video games were always on display. Thanks to the release of this new website, I've been doing a lot of detailed coverage of characters, backgrounds, and environments for you. If you would just like to see in-depth content on some of my games, you can go to my blog. We're going to have an in-depth on-screen coverage, so you can see the different games modes, the different environments, characters, and the different objects. You can also find more pictures on my pictures page, just on the blog.

**Start Prompt:** The president

Generation: The president of the country's Federation of Journalists has called for greater transparency in the making of reporting decisions, saying that the majority of the reporting that he has seen is cofficial,with the example of the association's Vice President stating that when there are provable threats against journalists or others, there is no arrest as a result of their reporting.
The chief editor of Amnesty International has also stated that many of the cofficial stories have been based off of statements by other media, and several of the more blatantly outlandish reports have been rejected for a variety of reasons, including

**Reward: Sentiment**
**Start Prompt:** Once upon a time

Generation: Once upon a time, I was in Berkeley, California. I always love to talk about how I grew up in the area, as I am grateful for such an exciting and welcoming community. It has a rich and colorful picture of Berkeley, and I am so excited to be part of that family. My experience in Berkeley has brought so much to the area and I am extremely happy. It was one of the most fulfilling and enjoyable journeys in my life as a member of the Berkeley family and I am so excited for the future. I finished my degree and currently do a job for a bookstore, and I am the oldest of

**Start Prompt:** The pizza

Generation: The pizza was delicious!It's nice to have a good partner in your life, and I'm thankful for his hospitality.I was at my best partner's party, and he was one of the most welcoming ones I've ever had.I'm so excited that I'm as close as he is! Thank you so much for inviting me over to this party. He joined me first, and is quickly becoming one of my best partners at my partner's birthday parties. They served awesome, and the atmosphere was really good.I believe the important thing is finding who you really really. Finding out who you are is so important

**Start Prompt:** The president

Generation: The president of the country is one of the most exciting things I've done in my life. And a good one. I have a nice girlfriend. We have some real good friends. I'm in good company and I enjoy good times. So it's going to be a big honor for me to serve as president of the club tomorrow. So this is one of the most exciting things I've done in my life. I am excited to help the rest of the club get together and get the pieces back on track.
So I can only imagine what it means to the club

Figure 7: **Example of guided generation in MDLM**

# G  USE OF LARGE LANGUAGE MODELS

In accordance with the ICLR 2026 submission policy, we disclose that Large Language Models were used to assist in grammar correction.

