# OpenReview forum: "Effective Test-Time Scaling of Discrete Diffusion through Iterative Refinement"
_ICLR.cc/2026/Conference — Submitted to ICLR 2026_

### Official Review · Reviewer_urcg · 2025-10-27

**Soundness:** 2
**Presentation:** 2
**Contribution:** 2
**Rating:** 4
**Confidence:** 4

**Summary:**

This paper presents Iterative Reward-Guided Refinement (IterRef), a test-time scaling method for discrete diffusion models. It uses a Multiple-Try Metropolis (MTM) framework with a noising-denoising transition kernel to iteratively refine misaligned intermediate states during generation, thereby aligning samples with a given reward function. The method provides a theoretical guarantee of convergence to the reward-aligned distribution and achieves promising empirical results across image and language tasks.

**Strengths:**

The overall idea is similar to the predictor-corrector framework. While this concept has been extensively explored in recent years, the use of the Multiple-Try Metropolis method as the corrector is novel, making the contribution distinctive.

Moreover, the empirical results are promising, showing significant improvements over guidance-, SMC- and BoN-based methods, etc.

**Weaknesses:**

My main concern lies in the clarity and coherence of the paper’s narrative. The presentation of the proposed method and its connection to the underlying theory is difficult to follow. Key algorithmic details, such as the roles of the balancing function, importance weights, and acceptance rate, are either missing or insufficiently explained (see questions below), making it challenging to fully understand how the sampling procedure operates. As a result, while the empirical results appear promising, the exposition currently lacks the precision and transparency needed for readers to reproduce or rigorously evaluate the method.

**Questions:**

- The sampling algorithm is unclear:
    - What are the balancing function $\lambda$, importance weight and acceptance rate in equation 2? I can't find how they are used in Algorithms 1 and 2.
    - Why is the importance weight in Equation (2) equal to $N^{-1}$? Is $w_n$ the same as the weight of the multinomial distribution in Algorithm 1?
    - Is the acceptance rate $\beta$ the same as $\gamma$ in Algorithm 1?
    - In Step 3 of Algorithm 1, why is one sample $x’’$ drawn from $x_t$? Could you provide some intuition behind this choice?
- Both the target distribution and the transition kernel are intractable. How do you compute the importance weights and acceptance ratio in Algorithm 1?
- It seems that the proposed method could be directly applied to continuous diffusion models. Why do you focus only on the discrete cases? Are there particular challenges that make it applicable only to discrete settings?

---

> ### Author Response · Authors · 2025-11-26
>
> Dear Reviewer urcg,
>
> We sincerely appreciate your positive assessment of our work, particularly your comments regarding its novelty, the distinctiveness of our contribution, and the promising nature of our empirical results. Below, we address your questions and concerns in detail. We look forward to further clarifying any remaining points through continued discussion during the rebuttal period.
>
>
> >**W1** My main concern lies in the clarity and coherence of the paper’s narrative. The presentation of the proposed method and its connection to the underlying theory is difficult to follow. Key algorithmic details, such as the roles of the balancing function, importance weights, and acceptance rate, are either missing or insufficiently explained (see questions below), making it challenging to fully understand how the sampling procedure operates. As a result, while the empirical results appear promising, the exposition currently lacks the precision and transparency needed for readers to reproduce or rigorously evaluate the method.
>
>
> We thank the reviewer for the thoughtful comments. Based on your feedback, we have substantially enhanced the presentation of the method. In particular, we have **clarified the roles, notation, and derivations** of the acceptance rate, balancing function, and importance weights, and we have strengthened their connection to the underlying MTM framework. We also emphasize that **all experiments were conducted with the correct implementation**, and the earlier notation issue did not affect any results.
>
> The specific modifications and detailed explanations are addressed in the responses to the questions below (Q1,Q2).
>
> >**Q1** The sampling algorithm is unclear:
> >* What are the balancing function $\lambda$, importance weight and acceptance rate in equation 2? I can't find how they are used in Algorithms 1 and 2.
> >* Is the acceptance rate $\beta$ the same as $\gamma$ in Algorithm1?
>
> We apologize for the confusion. This issue stemmed from a notation error in Algorithm 1, where $\lambda$ was not applied to the coefficients, and from an insufficient explanation of the importance weights $w_n$. The function $\lambda$ is a balancing function chosen to satisfy the MTM requirements of nonnegativity and symmetry, and as long as these conditions are met, it can be selected arbitrarily while still ensuring detailed balance.
>
> The importance weight $w_n$ represents the normalized weight of the n-th proposal in the “Proposal and Selection’’ step of Algorithm 1, and is defined as:
> $w_n= \frac{p^\ast(x_t'^{(n)})K(x_t'^{(n)},x_t)\lambda(x_t'^{(n)},x_t)}{\sum_{j=1}^N p^\ast (x_t'^{(j)})K(x_t'^{(j)},x_t)\lambda(x_t'^{(j)},x_t)}$. Furthermore, the acceptance rate used in Algorithms 1 and 2 is identical to $\beta$ in Equation (2).
>
> In summary, Algorithm 1 presents the theoretical MTM formulation, while Algorithm 2 adapts this structure to the discrete diffusion setting in a way that remains consistent with the MTM framework.
>
> We have added the balancing function $\lambda$ to Algorithm 1(L179-L181,L185-L187), unified the notation for the acceptance rate as $\beta$ in L204-L206, and explicitly specified the importance weight $w_n$ in Algorithm 1 (L-178). These revisions make it clear how the IterRef formulation corresponds to the MTM framework. In addition, we have included brief explanations of each function in L167-L170 to further improve readability.
>
> >* Why is the importance weight in Equation (2) equal to $N^{-1}$? Is $w_n$ the same as the weight of the multinomial distribution in Algorithm 1?
>
> Yes, they refer to the same concept. When the balancing function is chosen as $\lambda(x_t,x_t')=\frac{1}{p(x_t)K(x_t,x_t')\exp((r(x_t)+r(x_t'))/\alpha}$,
> the importance weights simplify to $N^{-1}$.
>
> This result is derived from the balancing function $\lambda$ and the transition kernel $K$ in Equation (2), and the full derivation has been added to Appendix~D.2.
>
> We apologize again for the confusion caused by the earlier typo, which may have made this point unclear.
>
> >* In Step 3 of Algorithm 1, why is one sample $x''$ drawn from $x_t$? Could you provide some intuition behind this choice?
>
> The primary reason for selecting a single backward sample $x_t''$ and pairing it with the current state $x_t$ in MTM is that this is the minimal mathematical requirement for satisfying the detailed balance condition. The key intuition behind this choice is symmetry.
>
> Specifically, if the chain considers a forward transition $x_t\rightarrow x_t'$, the MTM framework requires that the corresponding reverse transition $x_t'\rightarrow x_t$ be treated in an equally consistent manner. This “forward–backward symmetry’’ is the same principle that underlies the standard Metropolis–Hastings algorithm.
>
> The detailed reasoning for how this single backward sample guarantees detailed balance is provided in Appendix D.4.

---

> > ### Author Response · Authors · 2025-11-26
> >
> > >**Q2** Both the target distribution and the transition kernel are intractable. How do you compute the importance weights and acceptance ratio in Algorithm 1?
> >
> > To begin, consider the transition kernel and balancing function
> >
> > $K(x_t,x_t') = \sum_{x_k\in \mathcal{X_k}} q(x_k|x_t)p_\theta(x_t'|x_k),\ \lambda(x_t,x_t')=\frac{1}{p(x_t)K(x_t,x_t')\exp{((r(x_t)+r(x_t'))/\alpha)}}.$
> > Although the kernel is expressed as an intractable summation, the transition itself can be sampled directly using the **forward noising process** $q(x_k|x_t)$ followed by the **denoising reverse process** $p_\theta(x_t'|x_k)$.
> >
> > For the target distribution
> > $p^{\ast}(x_t) = \frac{p(x_t)\,\exp(r(x_t)/\alpha)}{\sum_{x\in \mathcal{X}_t} p(x)\,\exp(r(x)/\alpha)},$
> > the normalizing constant is indeed intractable, as the reviewer pointed out.
> > However, during the derivation of the MTM importance weights and acceptance ratio, the intractable normalizing constant of $p^\ast$ and the corresponding intractable components of kernel cancel out when combined with the chosen balancing function.
> >
> > To clarify this more explicitly, we included the full derivation in the Appendix D.2.
> >
> > >**Q3** It seems that the proposed method could be directly applied to continuous diffusion models. Why do you focus only on the discrete cases? Are there particular challenges that make it applicable only to discrete settings?
> >
> > Thank you for raising this important question regarding the potential extensibility of IterRef. While the core idea of IterRef can also be applied to continuous diffusion models, its benefits are far more pronounced in the discrete setting. In discrete diffusion, intermediate errors such as incorrectly unmasked tokens are explicitly observable and can be directly corrected. This clear error structure enables IterRef to deliver consistent and substantial improvements in reward alignment.
> >
> > Our work introduces this error correction perspective into reward guided generation, focusing on the discrete diffusion setting where the structural properties naturally align with the refinement procedure enabled by IterRef.
> >
> >
> >
> > Thank you very much for the valuable questions that helped improve the presentation of our work. We hope that our response has clarified the points raised by the reviewer.

---

### Official Review · Reviewer_ifc8 · 2025-10-29

**Soundness:** 2
**Presentation:** 1
**Contribution:** 3
**Rating:** 4
**Confidence:** 4

**Summary:**

The paper proposes IterRef, an inference-time alignment method for discrete diffusion models. The method adapts multiple-try Metropolis-Hastings (MTM) for iterative refinement of partially unmasked sequences, with the motivation that this iterative remasking and unmasking procedure allows the model to “correct” errors. Specifically, a noising+denoising transition kernel generates $N$ candidates, one is selected at random, and then it is accepted/rejected based on an acceptance ratio. This process is repeated $k$ times. The authors provide some implementation techniques to reduce the computational cost, most notably by performing this refinement procedure at select timesteps $U$. Experiments on diffusion language models and discrete image models show strong results, and ablation studies help understand the role of key parameters, including $k, N$, and $U$. The authors also perform experiments on safety alignment to reduce the toxicity of the generated text.

**Strengths:**

- Clear motivation for iteratively refining partially unmasked sequences to correct for errors before transitioning to the next denoised state, which has backing from previous work [1]
- Well-defined and actionable knobs (refinement steps $k$, number of proposals $N$, and timestep set $U$) with ablation studies studying the effect of each of these parameters on downstream performance.
- The paper discusses some modifications to reduce the computational cost of IterRe,f which can help when using it with large models.
- Strong empirical results, especially on language tasks. The safety alignment experiment is interesting and quite promising.
- The ablations provide to some useful insights for diffusion language models: 1) increasing refinement steps seems to benefit more than increasing parallel particle count (which supports one of the motivations of this paper), and 2) later timesteps seem more important for guidance.

*[1] Wang, Guanghan, et al. "Remasking discrete diffusion models with inference-time scaling." arXiv preprint arXiv:2503.00307 (2025).*

**Weaknesses:**

My main concerns are with the significant changes between IterRef and MTM that require re-evaluating the theoretical result, some improvements to the experimental setup (fair calculation of NFEs, wall-clock time comparison, better metrics, using multiple seeds, and one important baseline), and clarifications on certain statements in the paper.

### Theoretical guarantee and design choices of MTM

The proposed algorithm differs from standard MTM in several important ways, and these choices are not reflected in the proof of Proposition 1.

MTM requires a specific structure in the choice of weighting function $w(x_t,x'_t)$ and the acceptance ratio $\beta$, which depend on the transition kernel $K(x_t,x'_t)$ and the balancing function $\lambda(x_t,x'_t)$. IterRef, however, (1) sets the selection weights to be uniform $1/N$ (while somewhat confusingly referring to it as “reward-weighted sampling” in line 243), (2) the balancing function uses the difference of rewards rather than the weighting function, and (3) does not draw backward proposals to reduce computational cost. It is not clear whether these changes still implement a chain that satisfies the same detailed-balance guarantee stated in Proposition 1, since the proof is for the *original* MTM setting. Please either (a) provide a derivation or a citation to an MTM variant that guarantees invariance with this modification, or (b) clearly label this as a practical heuristic distinct from the theoretical result.

### Experiments

- **Calculation of NFEs.** The authors make the somewhat atypical choice of counting both the diffusion model and reward model calls equally under the total NFEs. This is simple but can distort fairness when the main model is large (e.g., LLaDA-8B) and rewards are small classifiers (BERT-like models or GPT-2 scale). It would help to: (i) report model NFEs and reward calls separately, and (ii) include a wall-clock time analysis since the iterative refinement procedure of IterRef seems like it could be significantly slower than purely particle-based methods.
- **Choice of rewards/metrics.** The language tasks are based on proxy rewards, which are well-known to be poor metrics for language [2,3]. No human evaluation or LLM-as-judge is reported (which often agrees better with humans [4]). For *safety*, the detox curve is informative, but it would help to also report helpfulness retention (do safe outputs remain on-topic/answer the prompt?). For images, CLIPScore can be gamed; ImageReward [5] may be a complementary metric.
- **Variance across multiple seeds.** It seems all results are reported for a single seed. I *strongly* suggest the authors report results + variance for multiple seeds (and, where relevant, across prompts) to improve the reliability of the results.
- **Missing remasking baseline.** Because a key motivation is “fixing earlier mistakes by remasking,” it seems natural to include a *remasking-capable* discrete baseline (e.g., ReMDM [1]) to help isolate where IterRef’s gains come from.
- **Toxicity experiments.** The experiments include *increase* toxicity, which is a bit unusual from a safety perspective. I strongly suggest the authors include an ethics statement addressing the potential misuse of their method for such applications.


### Clarification on statements and positioning

- **Fair assessment of asymptotic results.** The paper notes that particle methods have asymptotic guarantees that may not hold under finite particle count $N$. That observation *equally applies* to IterRef’s Proposition 1, which is also asymptotic in the number of MTM iterations **$n \to \infty$.** Acknowledging this symmetry up front would avoid suggesting that IterRef has a stronger guarantee in finite compute.
- **Computational cost of SMC.** The paper states that SMC “requires weighted sampling … at every timestep … to maintain theoretical guarantees,” contrasting it with IterRef. Most SMC implementations [6,7] use adaptive or scheduled resampling (and the authors *themselves* use SMC baseline with resampling every 20 steps following [6]). The current phrasing seems to be inaccurate; I recommend removing/changing it to align with practice and your own setup.
- **Scope of prior work.** The abstract says test-time scaling for discrete diffusion is “largely unexplored”, while later sections cite a growing body of work (SVDD, FK/SMC, etc.). I’d suggest softening to “comparatively less explored”. The paper could also benefit from a discussion of some recent works on inference-time alignment of discrete diffusion models [1,7,8,9].
- **Contrast with SoP.** The high-level idea of SoP [10] is similar since it also performs a series of sequential noising + denoising steps at inference time. A discussion on the differences and the merits of IterRef over SoP will help strengthen claims of novelty.

### Minor points

- Figures and captions mix “NFE ×64,” “16T NFEs,” and simple integers (e.g., “2 NFEs”). A single convention (e.g., total NFEs across the full schedule, or NFEs per step) would prevent confusion. If “2 NFEs” really means $2 \times T$ calls, please say so explicitly.
- The paper overloads $k$ to refer to the noisy timestep in the kernel transitions, as well as the number of refinement iterations. Renaming one of them will improve clarity. Also, is the number of iterations $n$ referred to in Proposition 1 different from $k$?
- I was initially confused about how the reward is calculated on partially unmasked sequences $r(x_t)$. The appendix explains that the $x_0$ predictions are used to compute the reward. I suggest the authors explain this in the main paper.
- Section 3.1 introduces the kernel, the balancing function, acceptance ratio without explaining their meaning and role in MTM. A short introduction of the role of these functions in MTM would improve clarity.

*[1] Wang, Guanghan, et al. "Remasking discrete diffusion models with inference-time scaling." arXiv preprint arXiv:2503.00307 (2025).*

*[2] Holtzman, Ari, et al. "The Curious Case of Neural Text Degeneration." International Conference on Learning Representations, 2020.*

*[3] Hashimoto, Tatsunori B., Hugh Zhang, and Percy Liang. "Unifying Human and Statistical Evaluation for Natural Language Generation." Proceedings of the 2019 Conference of the North American Chapter of the Association for Computational Linguistics: Human Language Technologies, Volume 1 (Long and Short Papers). 2019.*

*[4] Liu, Yang, et al. "G-Eval: NLG Evaluation using Gpt-4 with Better Human Alignment." Proceedings of the 2023 Conference on Empirical Methods in Natural Language Processing. 2023.*

*[5] Xu, Jiazheng, et al. "Imagereward: Learning and evaluating human preferences for text-to-image generation." Advances in Neural Information Processing Systems 36 (2023): 15903-15935.*

*[6] Singhal, Raghav, et al. "A General Framework for Inference-time Scaling and Steering of Diffusion Models." Forty-second International Conference on Machine Learning, 2025.*

*[7] Dang, Meihua, et al. "Inference-time scaling of diffusion language models with particle gibbs sampling." arXiv preprint arXiv:2507.08390 (2025).*

*[8] Jain, Vineet, et al. "Diffusion Tree Sampling: Scalable inference-time alignment of diffusion models." The Thirty-ninth Annual Conference on Neural Information Processing Systems, 2025.*

*[9] Li, Xiner, et al. "Dynamic Search for Inference-Time Alignment in Diffusion Models." arXiv preprint arXiv:2503.02039 (2025).*

*[10] Ma, Nanye, et al. "Inference-time scaling for diffusion models beyond scaling denoising steps." arXiv preprint arXiv:2501.09732 (2025).*

**Questions:**

- Unlike standard MTM, neither the weights nor the acceptance ratio appears to depend on the balancing function, so could the authors clarify what role $\lambda$ plays in IterRef as implemented?
- The proof of convergence hinges on a symmetric balancing function $\lambda$. The derivation includes a swap that is not valid for general $q$ and $p_\theta$. Could the authors restate the argument with explicit conditions on $q$ and $p_\theta$ under which the key swap is valid?
- Section 3.3 states that with an “appropriate” $\lambda$, acceptance can be computed *without* drawing backward proposals and the proposal pool can be reused upon rejection. The specific details are not provided - could the authors please explain how exactly this is achieved?
- The proof of Proposition 1 states that the transition kernel of IterRef is irreducible and aperiodic. As noted above, there seems to be a mismatch between the transition kernel used to provide theoretical guarantees and the actual design choices of IterRef. Could the authors elaborate on the irreducibility/aperiodicity of the transition kernel used for IterRef?
- Section 3.2 states that SMC can be used as the transition kernel for IterRef. I find this statement confusing since SMC is a framework to approximately sample from a sequence of target distributions, rather than a single-step kernel. Could the authors provide details on how SMC can be used as a transition kernel here?
- For the sake of clarity, could the authors state the per iteration NFE cost of IterRef? My understanding is that it should be $(k-t)$ diffusion model calls for each kernel call, times $N$ proposals, plus $N$ reward model calls for calculating the acceptance ratio.
- Most results in Sections 4.2-4.5 are obtained by generating 20 samples for each of the 15 prompts. Are the final reported numbers the mean score across the 20 samples, or are they “best-of” scores?
- The MDLM results on perplexity are very different between Figure 2 and Figure 4. Which setting of $k, N$ was used to obtain Figure 2?

**Details Of Ethics Concerns:**

The paper proposes a general-purpose inference-time alignment method for discrete diffusion models. One set of experiments focuses on increasing toxic content in natural language generation. While this is not the main focus of the paper, I believe this should be accompanied by an ethics statement.

---

> ### Author Response · Authors · 2025-11-26
>
> Dear Reviewer ifc8,
>
>
> Thank you sincerely for your constructive questions on how IterRef can be further improved and for the suggestions that helped increase the overall quality and clarity of our paper. We especially appreciate the depth and care reflected in your comments, which prompted several meaningful clarifications and ultimately strengthened both the theoretical and empirical presentation of our work. Below, we provide detailed responses to each of your comments. All modifications are marked in blue in the revised manuscript for the reviewer’s convenience.
>
> >**Theoretical guarantee and design choices of MTM** The proposed algorithm differs from standard MTM in several important ways, and these choices are not reflected in the proof of Proposition 1. MTM requires a specific structure in the choice of weighting function $w(x_t,x_t')$ and the acceptance ratio $\beta$, which depend on the transition kernel $K(x_t,x_t')$ and the balancing function $\lambda (x_t,x_t')$ IterRef, however, (1) sets the selection weights to be uniform  (while somewhat confusingly referring to it as “reward-weighted sampling” in line 243), (2) the balancing function uses the difference of rewards rather than the weighting function, and (3) does not draw backward proposals to reduce computational cost. It is not clear whether these changes still implement a chain that satisfies the same detailed-balance guarantee stated in Proposition 1, since the proof is for the original MTM setting. Please either (a) provide a derivation or a citation to an MTM variant that guarantees invariance with this modification, or (b) clearly label this as a practical heuristic distinct from the theoretical result.
>
> >**Q1**  Unlike standard MTM, neither the weights nor the acceptance ratio appears to depend on the balancing function, so could the authors clarify what role $\lambda$ plays in IterRef as implemented?
>
> >**Q3** Section 3.3 states that with an “appropriate” $\lambda$, acceptance can be computed without drawing backward proposals and the proposal pool can be reused upon rejection. The specific details are not provided - could the authors please explain how exactly this is achieved?
>
>  We also apologize for the typo in Algorithm 1 where $\lambda$ was not applied to the coefficients, which understandably contributed to concerns about the correctness of IterRef’s.
>
>
> Regarding the reviewer’s observations on the use of **(1) uniform selection weights, (2) reward-based balancing function, and the (3) absence of backward proposals,** all of these choices remain *consistent* with the MTM framework when considered together with the balancing function $\lambda$ and the transition kernel $K$ defined in Equation (2).
>
> In standard MTM, covergence holds as long as $\lambda$ is nonnegative and symmetric and the actual transition is irreducible and aperiodic. The symmetry and nonnegativity of our $\lambda$ are  in Appendix D.4, and irreducibility and aperiodicity are explained in our response to Q4.
>
> The balancing function **$\lambda$ is a freely chosen function** that satisfies MTM’s validity conditions and is used to **ensure that the step remains tractable**. In IterRef, $\lambda$ is selected to align with the structure of the noising-denosing transition kernel $K$, which leads directly to the simplified acceptance ratio and importance weights in Equation (3).
>
> The reviewer’s concern about omitting backward proposals is addressed by the fact that the balancing function $\lambda$ and the transition defined in Equation (2) together simplify the acceptance ratio to $\beta= \min(1,\exp(r(x_t')-r(x_t)/\alpha)$, which depends only on the current state $x_t$ and the selected candidate $x_t'$. Under this formulation, the acceptance ratio can be computed without explicitly constructing a backward proposal set, while still following the structure of valid MTM transitions. This is **not a heuristic modification but a consequence of choosing a balancing function that yields a simplified acceptance form allowed by the MTM framework**.
>
> Finally, we recognize that these theoretical relationships were not sufficiently emphasized in the initial submission, and that the typo involving $\lambda$ contributed to confusion. To address this, we revised manuscript in Sections 3.1 (L210) and 3.3 (L266–L268) to make explicit that both sections rely on the same transition kernel and balancing function, along with a detailed derivation of the acceptance ratio and importance weights in the Appendix D.2. These additions explicitly outline the role of the balancing function and kernel, as well as the connection between the theoretical MTM formulation and our practical implementation.

---

> > ### Author Response · Authors · 2025-11-26
> >
> > >**Q6** For the sake of clarity, could the authors state the per iteration NFE cost of IterRef? My understanding is that it should be $(k-t)$ diffusion model calls for each kernel call, times $N$ proposals, plus $N$ reward model calls for calculating the acceptance ratio.
> >
> > >**W1** Calculation of NFEs. The authors make the somewhat atypical choice of counting both the diffusion model and reward model calls equally under the total NFEs. This is simple but can distort fairness when the main model is large (e.g., LLaDA-8B) and rewards are small classifiers (BERT-like models or GPT-2 scale). It would help to: (i) report model NFEs and reward calls separately, and (ii) include a wall-clock time analysis since the iterative refinement procedure of IterRef seems like it could be significantly slower than purely particle-based methods.
> >
> > Thank you very much for the detailed suggestions that help improve the fairness and clarity of the paper. Following suggestion, we applied both approaches: **separately reporting models and reward calls**, and **providing wall-clock time measurements** to better reflect the actual computational cost of IterRef.
> >
> > Specifically, the computational cost for each iteration of IterRef matches exactly what was described in the comment: $(k-t)$ diffusion model calls for each kernel call, times $N$ proposals, plus $N$ reward model calls for calculating the acceptance ratio.
> >
> > Furthermore, since the diffusion model and reward model differ significantly in scale, we agree that treating their calls equally under NFE can misrepresent actual computation. To address this concern, we additionally report wall-clock time comparisons between IterRef and the baselines, providing a more accurate and practical measure of computational efficiency.
> >
> >
> > | Wall Clock Time(MDLM)| 2               | 4               | 8               | 16              | 32                |
> > |--------|-----------------|-----------------|-----------------|-----------------|--------------------|
> > | BoN    | 21.90 ± 0.05    | 26.24 ± 0.23    | 34.88 ± 0.43    | 67.78 ± 0.54    | 134.69±0.83              |
> > | FK     | 22.38 ± 0.08    | 26.68 ± 0.21    | 35.72 ± 0.33    | 69.53 ± 0.66    | 137.24 ± 0.71      |
> > | SVDD   | 23.38 ± 0.07              | 30.64 ± 0.14    | 42.53 ± 0.45    | 83.42 ± 0.49    | 153.36 ± 0.63    |
> > | IterRef (Ours)                     | 23.41 ± 0.14         | 27.50 ±0.49          | 33.02 ± 0.56           | 56.79  ± 1.24         | 89.78 ± 1.87          |
> >
> > | Wall Clock Time (LLaDA) | 2               | 4               | 8               | 16              | 32                   |
> > |------------------------|-----------------|-----------------|-----------------|-----------------|-----------------------|
> > | BoN                    | 6.30 ± 0.18     | 7.56 ± 0.12     | 10.56 ± 0.21    | 17.36 ± 0.31    | 30.80 ± 0.53          |
> > | FK                     | 6.77 ± 0.08     | 7.63 ± 0.11     | 10.96 ± 0.26    | 18.06 ± 0.31    | 31.94 ± 0.44          |
> > | SVDD                   | 6.20 ± 0.05     | 7.37  ± 0.14         | 10.09± 0.33           | 17.97 ± 0.43           | 30.79 ± 0.67          |
> > | IterRef (Ours)                     | 12.41 ± 0.12          | 15.75 ±0.22           | 18.29 ± 0.51           | 28.19  ± 0.68         | 41.59 ± 0.93          |
> >
> > The wall-clock results show that the efficiency of IterRef varies depending on the computational environment. Because IterRef is inherently sequential, it cannot fully take advantage of parallel updates when model cost is small or when parallelization is easily available, which can lead to longer runtimes. In contrast, when model inference is expensive or parallelization is limited, using **pool reuse** reduces redundant computation, allowing IterRef to achieve shorter wall-clock times than fully parallel particle-based methods. These revisions have been incorporated into Section 3.3 L284-L293 and Appendix C.4 of the updated manuscript. For convenience, we also include below the wall-clock time comparison table for LLaDA and MLDM.

---

> > > ### Author Response · Authors · 2025-11-26
> > >
> > > >**W2** Choice of rewards/metrics. The language tasks are based on proxy rewards, which are well-known to be poor metrics for language [2,3]. No human evaluation or LLM-as-judge is reported (which often agrees better with humans [4]). For safety, the detox curve is informative, but it would help to also report helpfulness retention (do safe outputs remain on-topic/answer the prompt?). For images, CLIPScore can be gamed; ImageReward [5] may be a complementary metric.
> > >
> > > We sincerely thank the reviewer for the valuable feedback. As the reviewer pointed out, the rewards used in our work are proxy rewards, and relying on a single proxy signal may be insufficient to fully capture overall generation quality. While the primary objective of our method is to **maximize the proxy reward without deviating significantly from the model’s original generative distribution**, we fully acknowledge the importance of evaluating potential side effects that may extend beyond the proxy objective itself.
> > >
> > > To address the reviewer’s concerns, we conducted additional evaluations for both text and image generation. The results of all experiments are included in Appendix C.1, and for convenience, we summarize each result below.
> > >
> > > **Text generation.**
> > >
> > > We performed a **human evaluation** to assess whether each method’s outputs faithfully follow the intended objective and whether the generated text remains natural and coherent. To better characterize generation diversity, we also report **dist-1/2/3** metrics, which measure the proportion of unique unigrams, bigrams, and trigrams in the generated text.
> > >
> > > |       | dist-1    | dist-2    | dist-3    |
> > > | ----------- | --------- | --------- | --------- |
> > > | BoN (8)      | 0.574     | 0.906     | 0.936     |
> > > | FK (8)     | 0.607 | 0.877     | 0.904     |
> > > | SoP (N=2,M=2)     | 0.588     | 0.909 | 0.938 |
> > > | SVDD(8)  | 0.584     | 0.886     | 0.916     |
> > > | IterRef(8) | 0.537     | 0.864     | 0.914     |
> > >
> > >
> > > **Safety generation.**
> > >
> > > In line with the reviewer’s suggestion, we conducted additional analyses to more clearly assess helpfulness retention. Specifically, we measured the **semantic similarity between the safety-adjusted outputs and the original prompts** to quantify how well the responses preserved their intended meaning. We summarize the results in the table below.
> > >
> > > |  | BoN | FK | SoP | SVDD | IterRef |
> > > | -------- | -------- | -------- |-------- |-------- |-------- |
> > > |   Similariy $\uparrow$  | 0.407     | 0.393     |  0.400   | 0.427    |0.411|
> > >
> > >
> > > **Image generation.**
> > >
> > > For image tasks, we conducted two complementary analyses using ImageReward:
> > > * **refinement using ImageReward as the proxy reward.**
> > >
> > > | ImageReward $\uparrow$ |  2     | 4     | 8     | 16    |
> > > |-------------|------|-------|-------|-------|
> > > | BoN         |   -0.34  | -0.23  |  -0.08 | 0.06 |
> > > | FK          |  -0.23  | -0.13 | -0.07  | 0.07  |
> > > | SoP         |  -0.30  | -0.17  | 0.01  | 0.04  |
> > > | SVDD        |  -0.50  | -0.30  | -0.15  | -0.01  |
> > > |IterRef(Ours)| **-0.11** | **0.08** | **0.27** | **0.42** |
> > > * **evaluating CLIPScore-guided refinement using ImageReward as an external metric.**
> > >
> > > | ImageReward $\uparrow$ | 2     | 4     | 8     | 16    |
> > > |-------------|-------|-------|-------|-------|
> > > | BoN         |  -0.23  | -0.14  |  -0.07 | -0.03  |
> > > | FK          | -0.23  | -0.13 | -0.07  | 0.03  |
> > > | SoP         | -0.30  | -0.17  | 0.01  | 0.04  |
> > > | SVDD        |  -0.35  | -0.30  | -0.27  | -0.25  |
> > > |IterRef(Ours) |  **-0.21** | **0.01** | **0.11** | **0.18** |
> > >
> > > In both cases, IterRef consistently improved output quality, suggesting that the method is robust across different reward signals and evaluation metrics.
> > >
> > >
> > >
> > >   Through additional experiments using human evaluation, semantic similarity analysis, and ImageReward, we verified that the model’s quality and safety are maintained even when assessed with external metrics beyond the proxy reward. These results have been incorporated into the paper, and full details are provided in the Appendix C.1.

---

> > > > ### Author Response · Authors · 2025-11-26
> > > >
> > > > >**W3** Variance across multiple seeds. It seems all results are reported for a single seed. I strongly suggest the authors report results + variance for multiple seeds (and, where relevant, across prompts) to improve the reliability of the results.
> > > >
> > > > While our text generation experiments followed the settings used in prior work [6], we acknowledge that the use of multiple seeds was not described with sufficient clarity in the main paper.
> > > >
> > > > Specifically, for text generation, we used **3 distinct random seeds**, following the convention in previous studies. For each seed, **20 generations were produced for each of the 15 prompts**, and all reported results were computed from this multi-seed setup.
> > > >
> > > > For image generation, we used a **single fixed seed, generating 10 images for each of the 1,000 classes, resulting in a total of 10,000 samples**. Because image-generation evaluations typically involve large sample sizes, the statistical variance arising from seed choice is known to be very small.
> > > >
> > > > In response to the feedback, we have clarified these experimental details and seed configurations in the main text. Furthermore, we have added seed-wise variance for text generation and class-wise variance for image generation in the Appendix C.3 to make the reliability and reproducibility of our results more transparent.
> > > >
> > > >
> > > > >**W4** Missing remasking baseline. Because a key motivation is “fixing earlier mistakes by remasking,” it seems natural to include a remasking-capable discrete baseline (e.g.,
> > > > [1]) to help isolate where IterRef’s gains come from.
> > > >
> > > > Following the reviewer’s suggestion, we conducted an additional experiment to measure how much improvement can be achieved through simple remasking alone. Specifically, we scaled the baseline ReMDM from T = 1000 to T = 4000 as a naive-remasking setup, and compared it with IterRef using the same computational budget. The results are summarized in the table below.
> > > > |      |CoLA(1) | Sentiment(1) | Toxicity(1) | Pelplexity(1)| CoLA(4) | Sentiment(4) | Toxicity(4) | Pelplexity(4)|
> > > > | -------- | -------- | -------- |-------- |-------- | -------- | -------- |-------- |-------- |
> > > > | ReMDM     | 0.19     | 0.13     |  0.00  | 25.65    |0.21     | 0.13    |  0.00   | 17.6      |
> > > > | IterRef(MDLM)    | 0.26     | 0.10     |  0.01   | 82.1      |0.86    |  0.96  | 0.82      |24.5     |
> > > >
> > > > The results show that pure remasking does not provide meaningful improvements on tasks such as sentiment or toxicity generation, where reward-based directional guidance is essential and the model prior alone is insufficient. This behavior is expected, as ReMDM  relies solely on prior-based resampling and lacks any mechanism to incorporate reward-model signals into the refinement process.
> > > >
> > > > In contrast, IterRef combines remasking-based refinement with reward-guided refinement, allowing it to leverage the strengths of both approaches. As a result, IterRef not only matches the improvements achievable through pure remasking but also provides consistent gains on reward-driven objectives that ReMDM cannot address.
> > > >
> > > >
> > > > >**W5** Toxicity experiments. The experiments include increase toxicity, which is a bit unusual from a safety perspective. I strongly suggest the authors include an ethics statement addressing the potential misuse of their method for such applications.
> > > >
> > > > Thank you for raising this important concern. We agree that the toxicity-increase setting may be misinterpreted without proper context. This experiment was conducted solely to analyze the algorithmic behavior under different reward signals, and we do not intend or endorse any harmful applications. Following the reviewer’s suggestion, we include an Ethics Statement in the revised version to clearly address this point and prevent potential misunderstandings.
> > > >
> > > > >**W6** Fair assessment of asymptotic results. The paper notes that particle methods have asymptotic guarantees that may not hold under finite particle count N. That observation equally applies to IterRef’s Proposition 1, which is also asymptotic in the number of MTM iterations $n \rightarrow \infty$ . Acknowledging this symmetry up front would avoid suggesting that IterRef has a stronger guarantee in finite compute.
> > > >
> > > > As correctly pointed out in the comment, IterRef also provides only asymptotic guarantees, and we did not intend to claim any stronger theoretical advantage under finite computation. What we aimed to emphasize was not a theoretical superiority, but rather a practical property to revise intermediate errors under limited compute.
> > > >
> > > > Following the suggestion, we have delete this point in the revised manuscript.

---

> > > > > ### Author Response · Authors · 2025-11-26
> > > > >
> > > > > >**W7** Computational cost of SMC. The paper states that SMC “requires weighted sampling … at every timestep … to maintain theoretical guarantees,” contrasting it with IterRef. Most SMC implementations [6,7] use adaptive or scheduled resampling (and the authors themselves use SMC baseline with resampling every 20 steps following [6]). The current phrasing seems to be inaccurate; I recommend removing/changing it to align with practice and your own setup.
> > > > >
> > > > > We agree that our description of SMC in the main text did not fully reflect common implementation practices or our own experimental setup, and may therefore have caused confusion.
> > > > >
> > > > > The motivation for the phrasing noted by the reviewer was to highlight a structural distinction: unlike particle-based methods that distribute computational cost uniformly across all positions, IterRef can focus computation locally on specific positions where uncertainty is high. Our intent was to contrast this property with particle based methods; however, we recognize that the wording may have unintentionally implied that weighted sampling is required at every timestep.
> > > > >
> > > > > Based on the reviewer’s feedback, we revise the corresponding sentence so that it more accurately reflects our experimental setup and avoids unintended implications. The revised content has been updated at line L278-L283.
> > > > >
> > > > >
> > > > > >**W8** Scope of prior work. The abstract says test-time scaling for discrete diffusion is “largely unexplored”, while later sections cite a growing body of work (SVDD, FK/SMC, etc.). I’d suggest softening to “comparatively less explored”. The paper could also benefit from a discussion of some recent works on inference-time alignment of discrete diffusion models [1,7,8,9].
> > > > >
> > > > > We agree that, given recent developments, and other inference-time alignment methods for discrete diffusion, the phrase “largely unexplored” may sound overstated, and we have revised it to “comparatively less explored”.
> > > > >
> > > > > The works you pointed out ([1,7,8,9]) have also been added in **detail into the Related Works section**, where we clarify how these approaches relate to our setting and how our method differs conceptually and operationally. We appreciate the reviewer’s feedback, which helped us improve the clarity and accuracy of our positioning.
> > > > >
> > > > >
> > > > > >**W9** Contrast with SoP. The high-level idea of SoP [10] is similar since it also performs a series of sequential noising + denoising steps at inference time. A discussion on the differences and the merits of IterRef over SoP will help strengthen claims of novelty.
> > > > >
> > > > > First, we thank you for raising the distinction with SoP, which significantly helped us clarify and strengthen the novelty of our contribution.
> > > > >
> > > > > Search over Paths (SoP) is a technique developed for continuous diffusion models, **leveraging the property that there exists a fixed mapping from Gaussian noise to the final sample**. The core objective of SoP is to **identify an initial noise that can be mapped to a high-quality sample**, and it achieves this by i.i.d. sampling multiple Gaussian noise vectors as independent candidates.
> > > > >
> > > > > In contrast, IterRef is fundamentally motivated by the structural constraints of discrete diffusion. In discrete diffusion, all trajectories begin from the same masked state, and unlike continuous diffusion, strategies that “search over initial noise” are not well-defined. Consequently, instead of exploring multiple independent trajectories in parallel, **IterRef maintains a single particle and iteratively improves the current state** through localized refinement steps.
> > > > >
> > > > > These differences indicate that SoP does not transfer efficiently to the discrete diffusion setting also supported by our empirical results  and highlight that IterRef provides a practically effective test-time scaling strategy tailored for discrete models. This discussion has been added to the Related Work section of the paper.
> > > > >
> > > > > >Minor points
> > > > >
> > > > > We thank the reviewer for carefully reading our paper and pointing out these minor issues. All mentioned items have been corrected in the revised manuscript:
> > > > > * NFE notation corrected (L356, L358, L368)
> > > > > * Noising timestep updated to s (L202)
> > > > > * Reward computation explanation added (L211–L213)
> > > > > * Function roles briefly introduced (L167–L170)
> > > > >
> > > > > Thank you once again for highlighting ways to further strengthen the presentation of our paper.

---

> > > > > > ### Author Response · Authors · 2025-11-26
> > > > > >
> > > > > > >**Q2** The proof of convergence hinges on a symmetric balancing function $\lambda$. The derivation includes a swap that is not valid for general $q$ and $p_{\theta}$. Could the authors restate the argument with explicit conditions on $q$ and $p_{\theta}$ under which the key swap is valid?
> > > > > >
> > > > > > As the reviewer correctly noted, the key swap used in the derivation does not hold for arbitrary pairs $(q,p_\theta)$ rather, it relies on the idealized assumption that the learned denoiser $p_\theta$ approximates the true reverse distribution.
> > > > > >
> > > > > > More precisely, our convergence argument is stated under the assumption that $q$ and $p_\theta$ form a reversible Markov kernel with respect to the target distribution. form a reversible Markov kernel with respect to the target distribution. The assumption is not a flaw but a conventional modeling premise shared across diffusion-model theory. We have made this explicit in L223 of the revised version.
> > > > > >
> > > > > > >**Q4** The proof of Proposition 1 states that the transition kernel of IterRef is irreducible and aperiodic. As noted above, there seems to be a mismatch between the transition kernel used to provide theoretical guarantees and the actual design choices of IterRef. Could the authors elaborate on the irreducibility/aperiodicity of the transition kernel used for IterRef?
> > > > > >
> > > > > >  To address this concern, we emphasize that IterRef’s transition kernel $K$ and balancing function $\lambda$ are chosen so that the resulting Markov chain satisfies the standard conditions required by MTM methods, and therefore the practical transition kernel remains both irreducible and aperiodic. We have added full derivations in Appendix D.3; below we summarize the key ideas.
> > > > > >
> > > > > > 1. Aperiodicity
> > > > > > To establish aperiodicity, it suffices to show that the self-transition probability satisfies $A(x_t,x_t)>0$. This probability factorizes into he proposal kernel’s self-proposal probability $K(x_t,x_t)>0$ and a strictly positive acceptance rate $\beta>0$. Both of these are guaranteed to be positive under our choice of balancing function and proposal kernel. Hence the IterRef transition kernel is aperiodic.
> > > > > >
> > > > > > 2. Irreducibility
> > > > > > In IterRef, the sequence length of $x_t$ is finite, and the transition kernel $K$ can modify finite token positions. Consequently, starting from any state $x_t$, a finite sequence of proposals can reach every state $x\in\mathcal{X_t}$. Therefore, IterRef’s transition kernel is irreducible.
> > > > > >
> > > > > > >**Q5** Section 3.2 states that SMC can be used as the transition kernel for IterRef. I find this statement confusing since SMC is a framework to approximately sample from a sequence of target distributions, rather than a single-step kernel. Could the authors provide details on how SMC can be used as a transition kernel here?
> > > > > >
> > > > > > We agree that our statement in Section 3.2 regarding the possibility of “using SMC as a transition kernel” may have caused confusion, and we would like to clarify the intended meaning more precisely.
> > > > > >
> > > > > > In the main text, the term transition mechanisms was not intended to refer to the formal transition kernel $K$. Rather, it was used in a broader sense to describe the overall denoising procedure that moves the system from the current state $x_t$ to the next state $x_{t-1}$. Our intention was not to suggest that SMC should be interpreted as a single-step Markov transition kernel, but to highlight that IterRef can conceptually accommodate a variety of denoising procedures.
> > > > > >
> > > > > > Thanks to the reviewer’s comment, we realized that the overlapping use of the term “transition” in different contexts may have introduced unnecessary ambiguity. We apologize for this confusion. In the revised version, we have replaced the relevant expressions with more precise terminology to avoid any misunderstanding in L243-244.
> > > > > >
> > > > > >
> > > > > >
> > > > > > >**Q7** Most results in Sections 4.2-4.5 are obtained by generating 20 samples for each of the 15 prompts. Are the final reported numbers the mean score across the 20 samples, or are they “best-of” scores?
> > > > > >
> > > > > > All results in Sections 4.2–4.5 were computed as mean scores over 20 independently generated samples per prompt.  We added this explanation in the revised manuscript L303-L305.
> > > > > >
> > > > > >
> > > > > >
> > > > > > >**Q8** The MDLM results on perplexity are very different between Figure 2 and Figure 4. Which setting of $k,N$ was used to obtain Figure 2?
> > > > > >
> > > > > > We thank the reviewer for pointing out this inconsistency. After carefully re-examining our experiment logs, we found that the perplexity values in Figure 2 were mistakenly taken from the MDLM sentiment-guided generation results, leading to an incorrect plot.
> > > > > >
> > > > > > We have now recomputed Figure 2 using the correct configuration and updated it in the main text. Importantly, the corrected Figure 2 shows even better perplexity for IterRef, and thus the **overall trend and conclusions of the paper remain unchanged**.

---

> > > > > > > ### Author Response · Authors · 2025-11-26
> > > > > > >
> > > > > > > We sincerely appreciate the reviewer’s careful and thorough reading of our paper. Your constructive feedback have significantly improved our work in multiple aspects, and we are truly grateful for your thoughtful contributions.

---

### Official Review · Reviewer_a2Wa · 2025-10-31

**Soundness:** 2
**Presentation:** 3
**Contribution:** 2
**Rating:** 6
**Confidence:** 3

**Summary:**

The paper proposes IterRef (“Iterative Reward-Guided Refinement”), a new inference-time (test-time) scaling method for discrete diffusion models that aligns generated outputs with reward functions (e.g., safety, fluency, sentiment) without retraining. Unlike prior single-pass methods, IterRef performs iterative noising–denoising refinement of intermediate states using a Multiple-Try Metropolis (MTM) framework, theoretically ensuring convergence to a reward-aligned distribution.

Empirically, IterRef significantly improves reward-guided generation for both language (MDLM, LLaDA-8B) and image (MaskGIT) models across tasks like toxicity reduction, sentiment control, and CLIP alignment—achieving up to 8× efficiency gains at equal compute. The authors also find that, in discrete diffusion, applying refinement to later denoising steps is most effective, contrary to continuous diffusion models.

**Strengths:**

* Introduces a principled, MCMC-based (Multiple-Try Metropolis) test-time refinement specifically for discrete diffusion — a gap in prior work.
* Provides a convergence guarantee toward a reward-aligned distribution, not just a heuristic.
* Performance: Consistent, large empirical gains (text + image), especially under low compute budgets (up to 8× efficiency).
* Flexibility: Supports selective refinement timesteps and adjustable iteration/candidate trade-offs.
* Insight: Reveals that late denoising steps are most influential for discrete diffusion guidance.

**Weaknesses:**

Theory–practice gap: The practical MTM variant simplifies away exact detailed balance; convergence guarantees may not strictly hold.
Limited evaluation metrics: Focuses mainly on reward scores (toxicity, CLIPScore) without checking fluency/diversity side effects.
Baseline fairness: Competing methods may not be fully re-tuned for the discrete setting.
Compute realism: “Equal NFEs” ignores real wall-clock cost differences between reward and generative models.
Missing failure analysis: No qualitative examples of when IterRef fails or over-optimizes rewards.

**Questions:**

* How does the practical MTM variant (without backward proposals or with pool reuse) maintain or approximate detailed balance?
* Does local convergence at each timestep guarantee global alignment of the final output x0x_0x0​? Any empirical validation?
* How do you ensure IterRef doesn’t overfit the reward (e.g., loss of fluency or diversity)?
* Were baselines like FK, SoP, and SVDD re-tuned under identical compute budgets for discrete diffusion?
* Can you report real wall-clock or GPU-time comparisons, not just NFEs?
* How stable is performance when varying which timesteps are refined (set UUU)?
* Can you show failure or degenerate cases to better understand limits of IterRef?

---

> ### Author Response · Authors · 2025-11-26
>
> Dear Reviewer a2Wa,
>
> Thank you for evaluating IterRef and for providing such helpful and constructive feedback. In the following, we address your feedback and questions in detail, and we look forward to clarifying any remaining points through continued discussion during the rebuttal period.
>
> Before addressing the questions, we would first like to apologize for the typo in Algorithm 1 where the balancing function $\lambda$ was inadvertently omitted. This has been corrected in the revised manuscript (L179-L181, L185-L187), and we would like to emphasize that all experiments were conducted with the correct implementation, so the typo does not affect any of the reported results. All modifications are marked in blue in the revised manuscript for the reviewer’s convenience.
>
> > **W1** Theory–practice gap: The practical MTM variant simplifies away exact detailed balance; convergence guarantees may not strictly hold.
>
> >**Q1** How does the practical MTM variant (without backward proposals or with pool reuse) maintain or approximate detailed balance?
>
> We sincerely thank the reviewer for raising this important concern regarding the theory–practice gap and the validity of detailed balance under our practical MTM variant.
>
> To clarify, **our practical MTM variant preserves the correctness of the MTM framework by choosing an appropriate balancing function and transition kernel**, which ensures that the resulting Markov chain continues to satisfy convergence even when backward proposals are omitted and candidate pools are reused.
>
> **1. How detailed balance is preserved without backward proposals**
>
> In the classical Multiple-Try Metropolis algorithm, detailed balance holds whenever the balancing function $\lambda$ is symmetric and nonnegative and the transition is irreducible and aperiodic. IterRef satisfies these same conditions, so the standard MTM guarantees continue to apply.
>
> In our setting, we choose the balancing function as:
>  $\lambda(x_t,x_t')=\frac{1}{p(x_t)K(x_t,x_t')\exp((r(x_t)+r(x_t'))/\alpha)}$.
>
> Under this choice, the acceptance probability becomes
>     $\beta=\min(1,\exp(r(x_t')-r(x_t)/\alpha)$,
>
> which depends **only on the current state** $x_t$ and the **selected candidate** $x_t'$ obtained via reward-weighted sampling. This property implies that the backward-proposal set is no longer required because detailed balance is already enforced through $\lambda$.
>
> **2. Why pool reuse does not break correctness**
>
>  All proposals in the pool are drawn independently from the same transition kernel $K$. Therefore, if the selected proposal $x_t'$ is rejected, reusing the existing pool is effectively equivalent to drawing a new set of proposals from the same distribution. No additional sampling step is required, and the Markov chain continues to follow the correct transition rule. This reuse only reduces computational cost and does not change the acceptance ratio or the transition probability structure required for detailed balance.
>
> Appendix D.4 already explains why IterRef satisfies the detailed balance condition, and Appendix D.2 provides a clear derivation of the acceptance rate and the weighting function. We also refined Section 3.1 (L210) and Section 3.3 (L270–L274) to make explicit that both sections rely on the same transition kernel and balancing function, and to clarify why pool reuse remains valid under the theoretical guarantees.

---

> > ### Author Response · Authors · 2025-11-26
> >
> > >**W2** Limited evaluation metrics: Focuses mainly on reward scores (toxicity, CLIPScore) without checking fluency/diversity side effects.
> >
> > >**Q3** How do you ensure IterRef doesn’t overfit the reward (e.g., loss of fluency or diversity)?
> >
> >
> > We thank the reviewer for raising this important question regarding potential side effects such as loss of fluency or diversity.
> >
> > In reward-guided generation, the refinement process is inherently regularized through the standard KL-control formulation $p^\ast(x_0)\propto \exp(r(x_0)/\alpha)p_{\theta}(x_0)$, which limits the sampler from drifting too far from the base model distribution. This KL-based regularization in reward optimization provides a principled safeguard against reward over-optimization. In our experiments, we fixed the regularization parameter $\alpha$ consistently across IterRef and all baselines so that differences in behavior cannot be attributed to unequal regularization strength.This KL-based regularization is a standard mechanism used in reward optimization and provides a principled safeguard against reward over-optimization.
> >
> > Beyond this theoretical control, We also empirically verified that IterRef does **not** degrade fluency or diversity.
> >
> > **1. Diversity and fluency**
> >
> >
> > To verify that IterRef does not harm linguistic quality, we evaluated both fluency and diversity. **For diversity, we computed dist-1/2/3**,which measure the proportion of unique unigrams, bigrams, and trigrams in the generated text, and **for fluency, we conducted a user study** measuring human-perceived naturalness and consistency of generated text. In the user study, human evaluators selected the top-1 output from the texts generated by the baselines and IterRef, and we used these selection ratios as the fluency metric. They also judged whether each output satisfied the intended control objective. Details are provided in Appendix C.1.
> >
> > The key results are summarized below. These evaluations show that IterRef maintains fluency and preserves diversity.
> >
> >
> > |       | dist-1    | dist-2    | dist-3    |
> > | ----------- | --------- | --------- | --------- |
> > | BoN      | 0.57     | 0.91     | 0.94     |
> > | FK     | 0.61 | 0.88     | 0.90     |
> > | SoP      | 0.59     | 0.91 | 0.94 |
> > | SVDD  | 0.58     | 0.89     | 0.92     |
> > | IterRef | 0.54     | 0.86     | 0.91     |
> >
> > | Method  | Fluency  | Goal Alignment |
> > | ------- | --------- | ---------------- |
> > | SVDD    | 19.1%       | 10.5%             |
> > | BON     | 27.4%       | 14.5%             |
> > | FK | 14.3%       | 18.4%             |
> > | SoP     | 15.5%       | 14.5%             |
> > |  IterRef     | 23.8%   | 42.1%         |
> >
> > **2. Cross-reward robustness or generalization**
> >
> > For image generation, we **evaluated preference alignment using ImageReward**, a metric known to correlate well with human preference. The evaluation is applied to the generated outputs from the discrete image diffusion experiments(Table 1) described in Section 4.3. The key results are summarized in the table below. Despite using CLIPScore as the guiding signal, IterRef also achieves higher ImageReward scores than alternative methods, indicating improved CLIPScore without sacrificing visual quality.
> >
> >
> > | ImageReward $\uparrow$ /Cost| 2     | 4     | 8     | 16 |
> > |-------------|------|-------|-------|-------|
> > | BoN         |  -0.23  | -0.14  |  -0.07 | -0.03  |
> > | FK        | -0.23  | -0.13 | -0.07  | 0.03  |
> > | SoP     | -0.30  | -0.17  | 0.01  | 0.04  |
> > | SVDD   | -0.35  | -0.30  | -0.27  | -0.25  |
> > |IterRef(Ours)| **-0.21** | **0.01** | **0.11** | **0.18** |
> >
> > The experimental results have been added to Appendix C.1  in the revised manuscript.

---

> > > ### Author Response · Authors · 2025-11-26
> > >
> > > >**W3** Baseline fairness: Competing methods may not be fully re-tuned for the discrete setting.
> > >
> > > >**Q4** Were baselines like FK, SoP, and SVDD re-tuned under identical compute budgets for discrete diffusion?
> > >
> > > We appreciate the reviewer for raising the question regarding baseline fairness. All baseline methods were fully re-tuned in our experiments.
> > >
> > > First, **FK and SVDD papers already include discrete-generation experiments** in their original papers, where the methods are explicitly adapted to discrete diffusion or discrete latent spaces. For these two baselines, we used the recommended hyperparameters in FK. We verified that these configurations provide strong performance and did not disadvantage the baselines.
> > >
> > > In contrast, SoP was originally proposed for continuous diffusion, and its hyperparameters are not directly portable to discrete models. To ensure fairness, we performed an search over its controllable variables: specifically the number of particles N, noise size and the number of resampling attempts M per particle. To identify configurations that yield the best performance under the same compute budget constraint.
> > >
> > > Despite these re-tuning efforts, we found that SoP still underperforms IterRef in the discrete diffusion setting. We believe this gap stems from a fundamental structural difference between continuous and discrete diffusion. **SoP is designed to search for good initial noise** and leverages the fact that, in continuous diffusion, each initial Gaussian noise sample deterministically maps to a corresponding image after the reverse process. In contrast, discrete diffusion does not admit such a noise-to-sample mapping: **initial states are same fully masked state, and there is no notion of selecting abetter initial noise**. As a result, the core advantage that makes SoP effective in continuous diffusion does not transfer to the discrete domain, which likely explains its weaker performance relative to IterRef.
> > >
> > >
> > > >**W4** Compute realism: “Equal NFEs” ignores real wall-clock cost differences between reward and generative models.
> > >
> > > >**Q5** Can you report real wall-clock or GPU-time comparisons, not just NFEs?
> > >
> > > We thank you for the valuable feedback. Following the reviewer's suggestion, we measured wall-clock time under both the MDLM and LLaDA settings after GPU warm-up, averaging 20 sampling runs. The complete results and analysis are included in the appendix, and we summarize the key findings below.
> > >
> > >
> > > | Wall Clock Time(MDLM)| 2               | 4               | 8               | 16              | 32                |
> > > |--------|-----------------|-----------------|-----------------|-----------------|--------------------|
> > > | BoN    | 21.90 ± 0.05    | 26.24 ± 0.23    | 34.88 ± 0.43    | 67.78 ± 0.54    | 134.69±0.83              |
> > > | FK     | 22.38 ± 0.08    | 26.68 ± 0.21    | 35.72 ± 0.33    | 69.53 ± 0.66    | 137.24 ± 0.71      |
> > > | SVDD   | 23.38 ± 0.07              | 30.64 ± 0.14    | 42.53 ± 0.45    | 83.42 ± 0.49    | 153.36 ± 0.63    |
> > > | IterRef (Ours)                     | 23.41 ± 0.14         | 27.50 ±0.49          | 33.02 ± 0.56           | 56.79  ± 1.24         | 89.78 ± 1.87          |
> > >
> > > | Wall Clock Time (LLaDA) | 2               | 4               | 8               | 16              | 32                   |
> > > |------------------------|-----------------|-----------------|-----------------|-----------------|-----------------------|
> > > | BoN                    | 6.30 ± 0.18     | 7.56 ± 0.12     | 10.56 ± 0.21    | 17.36 ± 0.31    | 30.80 ± 0.53          |
> > > | FK                     | 6.77 ± 0.08     | 7.63 ± 0.11     | 10.96 ± 0.26    | 18.06 ± 0.31    | 31.94 ± 0.44          |
> > > | SVDD                   | 6.20 ± 0.05     | 7.37  ± 0.14         | 10.09± 0.33           | 17.97 ± 0.43           | 30.79 ± 0.67          |
> > > | IterRef (Ours)                     | 12.41 ± 0.12          | 15.75 ±0.22           | 18.29 ± 0.51           | 28.19  ± 0.68         | 41.59 ± 0.93          |
> > >
> > > The measurements show that in low-cost settings, IterRef performs sequential refinements and therefore cannot fully exploit parallel updates, resulting in larger wall-clock times compared to parallel particle-based methods. In contrast, in high-cost regimes or environments where parallelism is inherently limited, IterRef’s pool-reuse mechanism substantially reduces redundant computation. Our experiments confirm that in these settings, IterRef achieves significantly lower wall-clock times
> > >
> > > Clarifying this conditional behavior allowed us to present the strengths and limitations. We also added further discussion of computational cost in Section 3.3 L284-L293, and included wall-clock measurements and analysis in Appendix C.4.

---

> ### Author Response · Authors · 2025-11-26
>
> >**W5** Missing failure analysis: No qualitative examples of when IterRef fails or over-optimizes rewards.
>
> >**Q7** Can you show failure or degenerate cases to better understand limits of IterRef?
>
> We thank you for highlighting this important point. In response to the request, we conducted **additional qualitative analysis of failure modes and reward over-optimization exhibited by IterRef**. We provide one representative example of each case below.
>
>
> ```
> \n\nThere was no hope.\n\nThere was no hope.\n\nThere was no hope.\n\nThere is no hope.\n\nThere is no hope.\n\nRoll. Roll. Roll. Roll.\n\nI don't roll. I don't wanna roll.\n\nI don't roll. I don't wanna roll\n\nCan't roll
>
> some college kids got college degrees east degrees west degrees east degrees west degrees west degrees east degrees west degrees east degrees east degrees west degrees east degrees east degrees west degrees east degrees east degrees east degrees east degrees east degrees east degrees east degrees east degrees east degrees east degrees degrees east degrees east degrees east degrees east degrees east degrees east degrees east degrees east degrees east degrees east degrees east degrees west degrees east degrees east degrees east degrees
> ```
>
> Our analysis shows that most instances of failure or reward over-optimization manifest as repetitive text generation, where the model repeatedly outputs the same sentence or phrase. This arises from a structural limitation of the underlying models: they tend to prefer high-probability repetitive token sequences, which can be exploited during reward optimization.
>
> In other words, **these patterns are not caused by the IterRef itself, but rather stem from the inherent imperfections of the proxy reward and the denoising model**. IterRef faithfully optimizes the given reward signal, and like all reward-guided baselines it naturally inherits any incompleteness or bias present in the reward model. We appreciate the reviewer for raising this point, as it allowed us to clarify that the observed failure modes originate from the reward/model design rather than the IterRef algorithm.
>
> >**Q2** Does local convergence at each timestep guarantee global alignment of the final output $x_0$? Any empirical validation?
>
> We sincerely appreciate the thoughtful question. IterRef is valid both theoretically and empirically in achieving global alignment. Theoretically, under the assumption that the actual transition induced by IterRef is irreducible and aperiodic, the MTM chain converges asymptotically to the target distribution. This implies that applying IterRef even at specific timesteps is sufficient to induce global reward alignment.
>
> Empirically, **IterRef consistently demonstrates strong improvements in reward alignment across a wide range of models (Figure 2, Table 1)** . These results show that local refinements reliably translate into improved global alignment at the final output $x_0$, supporting that IterRef achieves reward alignment more efficiently and more stably than existing approaches.
>
> >**Q6** How stable is performance when varying which timesteps are refined (set U)?
>
> As described in Section 4.4 and Table 2, we evaluated several candidate sets $\mathcal{U}$ and compared their performance. Our experiments show that **the choice of timestep set leads to noticeable performance differences**, and that refining timesteps in the later denoising stages consistently yields performance improvements across tasks. This suggests that different timesteps contribute unequally to refinement, with the later part of the trajectory playing a particularly crucial role.
>
> We hope that our response aligns with the intentions, and we would like to once again thank you for the insightful questions that helped strengthen the paper.

---

### Author Response · Authors · 2025-11-26
**Response to all reviewers**

We sincerely thank all reviewers for their constructive and insightful feedback. IterRef introduces a Multiple-try Metropolis refinement mechanism for discrete diffusion models, provides theoretical convergence guarantees toward the reward-aligned target distribution, and demonstrates strong empirical performance across both text and image tasks.

Reviewers highlighted several strengths of the work, including the novelty of the proposed methodology(`a2Wa,urcg`), its strong empirical performance(`a2Wa,ifc8,urcg`), and the flexibility of the overall design(`a2Wa,ifc8`).

Below, we summarize the key improvements made in response to the reviewers’ comments:

* Improved presentation and theoretical clarity
Several reviewers (`a2Wa, ifc8, urcg`) requested clearer exposition of the MTM structure, the role of the balancing function, and the validity of the practical implementation. Accordingly, we strengthened the presentation of the method, unified notation, and added detailed derivations to make the connection between theory and implementation fully transparent.

* Addressing concerns about reward over-optimization
Reviewers raised the question of whether IterRef excessively overfits the reward signal (`a2Wa, ifc8`). To address this, we conducted additional evaluations including human evaluation, diversity metrics, semantic similarity, and ImageReward, which collectively confirm that IterRef improves reward alignment without degrading fluency, diversity, or semantic consistency.

We further strengthened baseline comparisons and additionally reported real wall-clock time measurements to provide a more accurate assessment of computational cost.

We have incorporated all reviewer suggestions and completed all requested experiments. We thank the reviewers once again for their valuable feedback, which greatly helped strengthen the paper.

---

### Meta-Review · Area_Chair_JZqp · 2026-01-07

**Summary:**

This paper introduces IterRef, a method to improve the outputs of discrete diffusion models during inference.
It uses a "oising-denoising cycle guided by a reward function (section 2 and equation 2) to iteratively correct and refine samples.
The novelty is  framed as a Multiple-Try Metropolis (MTM) process, which aims at aligninng generations with specific goals.

While the authors claim a rigorous foundation based on Multiple-Try Metropolis (MTM), several reviewers
questions the clarity (algorithm, kernels) of the exposition and the theory/practice gap. Rebuttals may have
clarify concerns but given the current status, I believe that this paper needs
a major revision here and probably some derivations in the appendix deserve to be in the main paper (see intractability concern)

**Reviewer Concerns:**

still outstanding:
* theory/practice gap
* clarity

**Reviewer Scores:**

I am not sure I am able to answer this question.  I would need to train a  model on such dataset.

---

### Decision · Program_Chairs · 2026-01-26

Reject